# Source apportionment of volatile organic compounds in the north-west Indo-Gangetic Plain using positive matrix factorisation model

**Pallavi** [1]**, Baerbel Sinha**[1]**, and Vinayak Sinha**[1]

[1]Department of Earth and Environmental Sciences, Indian Institute of Science Education and Research Mohali, Sector 81, S.A.S Nagar, Manauli PO, Punjab, 140306, India.

**Correspondence:** Baerbel Sinha (bsinha@iisermohali.ac.in)

**Abstract.** In this study we undertook quantitative source apportionment for 32 volatile organic compounds (VOCs) measured at a suburban site in the densely populated North-West Indo-Gangetic Plain using the US EPA PMF 5.0 Model. Six sources were resolved by the PMF model. In descending order of their contribution to the total VOC burden these are "biofuel use and waste disposal" (23.2%), "wheat-residue burning"(22.4%), "cars" (16.2%), "mixed daytime sources"(15.7%) "industrial emissions and solvent use"(11.8%) and "two-wheelers" (8.6%).

Wheat residue burning is the largest contributor to the total ozone formation potential (32.4%). For the emerging contaminant isocyanic acid, photochemical formation from precursors (37%) and wheat residue burning (25%) were the largest contributors to human exposure. Wheat residue burning was also the single largest source of the photochemical precursors of isocyanic acid, namely, formamide, acetamide and propanamide, indicating that this source must be most urgently targeted to reduce human concentration exposure to isocyanic acid in the month of May. Our results highlight that for accurate air quality forecasting and modelling it is essential that emissions are attributed only to the months in which the activity actually occurs. This is important both for emissions from crop residue burning (which occur in May and from Mid-October to the end of November).

The SOA formation potential is dominated by cars (36.9%) and two-wheelers (21.1%), which also jointly account for 47% of the human class I carcinogen benzene in the PMF model. This stands in stark contrast to various emission inventories which estimate only a minor contribution of the transport sector to the benzene exposure (∼10%) and consider residential biofuel use, agricultural residue burning and industries to be more important benzene sources. Overall it appears that none of the emission inventories represent the regional emissions in an ideal manner. Our PMF solution suggests that transport sector emissions may be underestimated by GAINSv5.0 and EDGARv4.3.2 and overestimated by REASv2.1, while the combined effect of residential biofuel use and waste disposal emissions as well as the VOC burden associated with solvent use and industrial sources may be overestimated by all emission inventories. The agricultural waste burning emissions of some of the detected compound groups (ketones, aldehydes and acids) appear to be missing in the EDGARv4.3.2 inventory.

## 1 Introduction

Volatile organic compounds (VOCs) have diverse natural $(760\,\mathrm{Tg(C)\,y^{-1}}$ (Sindelarova et al., 2014)) and anthropogenic sources $(127\,\mathrm{Tg\,y^{-1}}$ average value (IPCC, 2013)). Certain VOCs emitted primarily by anthropogenic sources such as benzene and isocyanic acid, have direct adverse impacts on human health even at low ppb concentration exposures (Chandra and Sinha, 2016). In densely populated regions like the Indo Gangetic Plain (IGP), reactive anthropogenic VOCs contribute significantly towards the formation of health relevant secondary pollutants such as ozone and secondary organic aerosol (Chandra and Sinha, 2016; Sarkar et al., 2016). At our study site, a representative suburban site in the NW-IGP, the 8 h average NAAQS (national ambient air quality standard) for ozone limit of 100 $\mu\mathrm{g\,m^{-3}}$ was exceeded on 29 out of 31 days during May 2012 (Sinha et al., 2014), while the 24 h average NAAQS for $PM_{2.5}$ of 60 $\mu\mathrm{g\,m^{-3}}$ was exceeded during 27 out of 31 days in the same

period. It has been shown that wheat residue burning results in significant enhancement (by 19 ppb) of the daytime ozone mixing ratios in pre-monsoon season (Kumar et al., 2016) and long range transport in the form of dust storms from the Arabian Peninsula brings extremely high $PM_{2.5}$ mass loadings (with peak $PM_{2.5}$ mass loadings of 950 μg m$^{-3}$ on $17^{th}$ of May 2012) (Sinha et al., 2014; Pawar et al., 2015) and enhances the $PM_{2.5}$ mass by $\sim 30\%$.

However, ozone mixing ratios exceed the NAAQS even during the non-fire influenced days of the pre-monsoon season and the NAAQS for $PM_{2.5}$ is exceeded 60% of the time for air masses with no history of long range transport (Kumar et al., 2016; Pawar et al., 2015). This indicates that local ozone and $PM_{2.5}$ precursor emissions deserve further study.

Previous source receptor modelling studies of VOC emission from India (Srivastava, 2004; Srivastava et al., 2005; Majumdar et al., 2009) produced results that conflicted strongly with the bottom up emission inventories, all of which contain significant emissions from residential fuel usage even when filtered for the New Delhi National Capital Region (41-45%), Greater Mumbai (32-36%) and Greater Kolkatta (33-59%). Transport sector emissions, according the bottom up emission inventories contribute only 15-35%, 17-43% and 6-14% to the total VOC emissions in New Delhi National Capital Region, Greater Mumbai and Greater Kolkatta, respectively. All previous studies employed a chemical mass balance (CMB) technique for ambient VOC source attribution and identified the transport sector as the main source in the form of evaporative emissions (40-87%) in Mumbai (Srivastava, 2004)), diesel internal combustion engines (26-58%) in Delhi (Srivastava et al., 2005) and roadway/refuelling exhaust (40%) in Kolkata city (Majumdar et al., 2009). Except for the study performed in Kolkata which found a contribution of <10% from wood combustion, residential fuel usage was not identified as a potential VOC source in those source receptor modelling studies. The observed discrepancy could be partially caused by the fact a CMB is not necessarily an ideal tool for conducting source receptor modelling study in understudied environments as the model needs to be initialized with locally measured source profiles of all potentially significant sources. However, it is unlikely that this is the only reason for the discrepancies between source receptor modelling outcomes and emission inventories. The only other source receptor modelling study in South Asia was conducted using a positive matrix factorisation model (EPA PMF5.0) with data collected in the Kathmandu valley, Nepal, as part of the SUSKAT campaign and attributed a negligible fraction of the anthropogenic VOC burden to residential biofuel usage ( 14%). Instead different industrial sources including brick kilns (jointly 52%) and the transport sector (21%) were identified as the dominant VOC sources in the Kathmandu valley.

Different bottom up emission inventories have large discrepancies between each other when extracted for the NW-IGP. For our study region (27.4-34.9 °N and 72-79.8 °E), EDGAR v4.3.2 (Huang et al., 2017) estimates that the road transport sector contributes only 18% of the total anthropogenic VOC emissions (440 Gg y$^{-1}$), while REAS v2.1 (Kurokawa et al., 2013) attribute 35.8% of the total anthropogenic VOC emissions (1227 Gg y$^{-1}$) to this sector. For industrial emissions and solvent use, GAINS (Amann et al., 2011) has the lowest (540 Gg y$^{-1}$) and EDGAR v4.3.2 the highest absolute emissions (900 Gg y$^{-1}$). Crop residue burning as VOC source is missing in REAS but accounted for a 6% (145 Gg y$^{-1}$) and 7% (163 Gg y$^{-1}$) share of the annual VOC emissions in EDGARv4.3.2 and GAINS, respectively. Considering the large discrepancies between bottom up inventories and different source receptor modelling studies, more source receptor modelling studies using robust statistical tools and better tracers for different biomass burning sources are necessary.

In the present study, we applied the US EPA's PMF 5.0 model in constrained mode for source apportionment of 32 VOCs measured at IISER Mohali Atmospheric Chemistry Facility in May 2012 with the objective of quantifying the most important sources of ozone and SOA precursors, the human class I carcinogen benzene and the emerging contaminant isocyanic acid (Chandra and Sinha, 2016), so that strategies for air pollution mitigation can benefit from quantitative evidence concerning the contribution of major sources. The month of May is of special interest, as it is affected by widespread wheat residue burning in the NW-IGP. In the present study, we quantify the contribution of this important area source to the VOC burden at a downwind site. Our analysis includes several rarely reported nitrogen containing compounds which appear to have strong pyrogenic sources in this particular study region. Compounds such as amines, amides and isocyanic acid are presently not included in global emission inventories and the default atmospheric chemistry mechanisms, despite their potential importance for secondary aerosol formation and human health. We compare our source-receptor modelling output with several emission inventories such as REAS v2.1, EDGAR v4.3.2 and GAINS v5 to assess which emission inventory is most consistent with the results of our source receptor modelling study that employs in-situ observations.

## 2  Methods

### 2.1  Receptor site

The measurement facility is situated inside Indian Institute for Science Education and Research Mohali (IISER Mohali) campus (Figure 1a) which is a suburban site (30.667 °N, 76.729 °E, 310 m above mean sea level) in Mohali near Chandigarh in India (Figure 1b). Collectively the metropolitan of Chandigarh-Mohali-Panchkula forms a tri-city with a total population of 1,941,118 (Census, 2011). The main air

transport toward the site was from the North West and the period studied was impacted by wheat residue burning, a dust storm and strong photochemistry (Sinha et al., 2014). Figure 1a shows 72 h HYSPLIT back trajectories arriving at the site. With average wind speeds of 5.6 m s$^{-1}$ during the study period (range 1-20 m s$^{-1}$) the meteorological conditions were conducive for capturing the contribution of regional emission sources. The measurement site, the meteorology and the primary dataset acquired during May 2012 have been discussed in detail elsewhere (Sinha et al., 2014).

## 2.2 VOCs and other Auxiliary measurements

We used hourly data of 32 measured organic ions which were assigned to volatile organic compounds (Supplementary Table S1) based on PTR-TOF-MS studies conducted by our group within the South Asian environment (Sarkar et al., 2016) to initialize the US EPA PMF 5.0 model and employed $CO$, $SO_2$, $O_3$ and $NO_y$ as independent tracers to validate the results. As described in greater detail in (Sinha et al., 2014), ambient air sampling was performed continuously through a Teflon inlet line protected by an in-line Teflon filter. A high sensitivity proton transfer reaction quadrupole mass spectrometer PTR-QMS (HS Model 11-07HS-088, Ionicon Analytik Gesellschaft, Austria) was operated at drift tube pressure of 2.2 mbar, a drift tube temperature of 60 °C and a drift tube voltage of 600 V, which resulted in an operating E/N ratio of ~ 135. Carbon monoxide (CO), Sulphur dioxide ($SO_2$), Ozone ($O_3$) and $NO_y$ (NO, $NO_2$ and other nitrogen species converted to NO by a molybdenum converter such as nitric acid and PAN) were measured using Thermo Fischer Scientific 48i (IR filter correlation based spectroscopy), 43i (pulsed UV fluorescence), 49i (UV absorption photometry) and 42i trace level air quality analysers (chemiluminescence), respectively.

## 2.3 Positive Matrix Factorisation model

In the current study, US EPA PMF 5.0 receptor model (Norris et al., 2014) was applied to the ambient VOC dataset (in µg m$^{-3}$) from May 2012 measured at the IISER-Mohali Atmospheric chemistry facility comprising of data matrix of 721 samples (rows) and 32 species (columns). The EPA PMF 5.0 receptor model (Paatero et al., 2014; Norris et al., 2014)is multivariate factor analysis tool (Paatero and Tapper, 1994; Paatero, 1997), which decomposes the data matrix x$_{ij}$ with $i$ number of samples and $j$ number of measured VOCs into two matrices, the factor contribution matrix g$_{ik}$ (which provides the mass g contributed by each factor to the individual sample) and the factor profiles matrix f$_{kj}$ (which provides the source profile/fingerprint of each individual source). Both matrices are established for a user defined number of sources $p$ from the existing intrinsic variability in the dataset leaving behind a matrix of residuals e$_{ij}$.

$$X_{ij} = \sum_{k=1}^{p} g_{ik}f_{kj} + e_{ij} \qquad (1)$$

A detailed description of the model can be found elsewhere (Paatero and Tapper, 1994; Paatero, 1997; Paatero et al., 2014; Norris et al., 2014). The two primary advantages of the PMF over other source receptor modelling tools are its inherent non-negative constraints (Hopke, 2016) and its capability of optimally weighing individual data points and assigning uncertainties which makes it possible to include less robust species that can be useful for defining real sources. The EPAv5.0 model is superior when compared to other source receptor modelling tools as contains advanced rotational features (Paatero and Hopke, 2009) which allow to constrain the rotational ambiguity in a manner that pushes the PMF solution toward the real world space.

All 32 species were assigned a fixed 20 % in the uncertainty , which represents the largest uncertainty estimated for strong compounds, to avoid a situation where the difference in the assigned uncertainty drives the PMF to dedicate a separate factor towards minimizing Q of a single compound with low uncertainty (toluene) by taking it out of all other source profiles and opening a separate factor profile containing just a single compound. 18 were identified as weak based on the signal to noise ratio and the presence of potential isobaric interferences as detailed in table S2. For weak species, the PMF model triples the stated uncertainty to reduce their impact on the models solution.Designating sources with isobaric interferences as weak is warranted, because when two sources with different temporal profiles (night-time combustion and daytime biogenic emission or night-time combustion and daytime photochemistry) could potentially contribute different compounds to the same m/z ratio, zero values are almost absent in that particular column of the matrix and the tracer is affected by additional uncertainty not appropriately expressed by merely looking at the instrumental measurement error and the signal to noise ratio. When this column is made "weak" and given a higher uncertainty, other "strong" tracers, representing a single compound, define source profiles and this reduces the rotational ambiguity of the model. The "weak" compounds with isobaric interferences tend to be distributed among the source profiles available as per the solution that minimizes Q but they do not define any of the profiles. The extra modelling uncertainty was kept to zero and missing values ($< 5\%$) were excluded. For every base model run, we used 20 runs with random seeds. Stable Q-values were obtained for all runs. The model was run with 3 to 7 factors, to identify the appropriate number of factors as discussed in the supplementary text in greater detail. Figure 2 shows the percentage contribution of the identified sources to the VOC burden for these runs. Figures S4 a, b and c show how the factor profile, percentage of each VOC originating from a certain source, and the factor con-

tribution change while increasing the number of factors in the model. Figure 2 shows that a 7 Factor solution provides little advantage over a 6 Factor solution while a 5 Factor solution does not resolve the wheat residue burning source which is independently verified by MODIS (Moderate Resolution Imaging Spectroradiometer) fire counts over the region. The residuals for all species for the 6 Factor solution depicted a normal curve and fall within -3.3 sigma and +3.3 sigma for all species indicating a good model fit. The constraints feature of the 5.0 version of the model was utilised to improve the performance of the model further as described in greater detail in the supplementary text. The constrained model operation of the PMF version 5.0 allows to reduce the rotational ambiguity of the model using external knowledge. For example, if a source is inactive for a particular period (as is photochemistry at night), then the source contribution ($g_{ik}$) due to that factor during that time period can be pulled to zero in the model to provide more robust output. Similarly, a compound that is known to be present only in primary emissions can be pulled down in the source composition ($f_{kj}$) matrix of the photochemistry factor. A list of the constraints applied is provided in the supplementary table S3. A detailed discussion of the use of constraints in a receptor model has been provided in previous studies (Paatero et al., 2002, 2014; Paatero and Hopke, 2009; Norris et al., 2014; Sarkar et al., 2016). Bootstrap model runs (Brown et al., 2015) were performed to assess the model uncertainty. Input parameters for the bootstrap runs constituted random seed, 100 number of bootstraps and default values for block size (10) and minimum correlation R-value (0.6) and there were no unmapped factors. Except for the car and two-wheeler factor (R=0.6) for which a certain degree of co-linearity is expected, none of the other factors showed cross correlation with each other (R<0.3) and the g-space plot even of this factor pair is well filled. The constraint mode was unable to force the PMF model to separate the wheat residue burning factor in a 5-factor solution without imposing a split between the car and 2-wheeler factor, indicating that these two indeed represent distinct source profiles.

## 2.4    Validation of the PMF output

The PMF generates two matrices from the intrinsic variability in the dataset. A factor contribution matrix and a factor profile matrix.

Traditionally the PMF output has been validated by cross-correlating the factor contribution matrix with independent tracers which were not used to initialize the model, but are considered useful tracers for the respective source (Brown et al., 2015; Leuchner and Rappenglück, 2010; Gaimoz et al., 2011; Bon et al., 2011; Sarkar et al., 2016). We perform this validation step for all six source factors resolved by the PMF model. These were identified as "biofuel use and waste disposal", "wheat-residue burning", "four-wheelers", "two-wheelers", "industrial emissions and solvent use" and

"mixed daytime sources", respectively. The factor contribution for 4-wheelers (R=0.7) and 2-wheelers (R=0.6) correlated best with the independent tracer $NO_y$ which is considered to be a vehicular exhaust marker (Ramanathan et al., 1985). The factor contribution of the domestic fuel usage and waste disposal factor correlated best with the independent tracer CO (R=0.9), a proxy for inefficient combustion, while the factor contribution of the industrial emission factor correlated best with the independent tracer $SO_2$ (R=0.6). The wheat residue burning factor days showed a moderate cross correlation with MODIS fire counts with an R=0.4 and a lag of 2 days. $O_3$ (R=0.8) was the best independent tracer for the mixed daytime factor.

However, our study goes one step further than all previous studies in validating the PMF output. For 5 out of 6 factors we validated the factor profiles generated by the PMF model against grab samples collected at the source. Factor profiles were cross-correlated with the fingerprints of source samples collected from a number of potential sources including wheat residue fires (Chandra et al., 2017; Kumar et al., 2018), a ambient air samples from a busy traffic junction (Chandra et al., 2017)and an industrial area (this study), tailpipes of various vehicles (this study), waste burning (Sharma et al., 2019), leaf litter burning (this study), domestic biofuel use (Stockwell et al., 2016) and brick kilns (Zhong et al., 2019) to identify the sources. Figure 3 shows the factor profiles obtained from the PMF run (in dark blue), the percentage of each species explained by the respective PMF factor (red squares) and the source profiles of those sources which best matched the factor profile (in various colors as indicated in the legend). The factor profile of residential fuel usage and waste disposal correlates most strongly with the measured VOC source speciation profiles of domestic cooking (R=0.8), leaf-litter burning (R=0.7) and smoldering garbage fires (R=0.6), the wheat residue burning factor with flaming wheat residue burning (R=0.9), the 4-wheeler factor with the tailpipe exhaust of petrol-fueled cars (R=0.5), gasoline evaporation headspace for diesel (R=0.5) and urban traffic junction grab samples (R=0.8) and the 2-wheeler factor with the tailpipe exhaust of petrol-fuelled 4-stroke two-wheelers (R=0.6). The industrial emissions correlated best with the source profile of brick kilns (R=0.5) and ambient air samples collected in an industrial area (0.6). For mixed daytime sources no source profile sampling is possible.

## 2.5    Conditional Probability Function analysis

We perform a conditional probability function (CPF) analysis (Leuchner and Rappenglück, 2010) which aids in identifying physical locations of different PMF source factors without using back trajectories (Xie and Berkowitz, 2006) The CPF is computed using the factor contribution of the PMF model in combination with the wind direction at the receptor site. It quantifies the probability of factor contributions surpassing a certain threshold ($75^{th}$ percentile) for a

particular wind direction sector thereby highlighting directional dependency of source factors and is defined as follows:

$$CPF = \frac{m_{\Delta\theta}}{n_{\Delta\theta}} \qquad (2)$$

Where $m_{\Delta\theta}$ represents the number of data points in the wind direction bin $\Delta\theta$ which exceeded the threshold criterion and $n_{\Delta\theta}$ represents the total number of data points from the same wind direction bin. $\Delta\theta$ was assigned a value of $30°$.

## 2.6 Calculation of the ozone formation potential and SOA formation potential

Ozone production potential ($O_3PP$) for each of the PMF derived source factors was calculated based on the method used by Sinha and co-workers (Sinha et al., 2012) using the following equation:

$$O_3PP = \left(\sum_i k_{VOC_i+OH}[VOC_i]\right) \times [OH] \times n \qquad (3)$$

using n = 2 and $[OH] = 10^6$ molecules cm$^{-3}$. The values were summed up for all the VOCs for obtaining the ozone production potential corresponding to each of the PMF derived factors for the daytime hours (07:00-18:00 LT).

Secondary organic aerosol (SOA) potential was calculated for the PMF source factors using the literature SOA yields (Derwent et al., 2010) under low $NO_X$ conditions for benzene, toluene, ethylbenzene, trimethylbenzene, styrene, methanol, isoprene, formaldehyde, acetaldehyde, acetone, formic acid and acetic acid using the equation given below for 07:00-18:00 LT.

$$SOApotential = \left(\sum_i [VOC_i]\right) \times [SOA_i] \qquad (4)$$

## 2.7 Methodology for the comparison of PMF source factors with existing emission inventories

Global Emission Database for Global Atmospheric Research (EDGARv4.3.2) inventory for the year 2012 (Huang et al., 2017) and two regional emission inventories: Regional Emission inventory in Asia (REAS v2.1) for the year 2008 (Kurokawa et al., 2013) and the Greenhouse Gas and Air Pollution Interactions and Synergies model (GAINS) (Amann et al., 2011) for the year 2010 (Stohl et al., 2015) were compared with our PMF output. The gridded inventory was filtered for Latitude: 27.4-34.9 °N and Longitude: 72-79.8 °E , i.e. the fetch region from which the air mass trajectories reach the receptor site within one day. This filtering is required because compounds with photochemical lifetimes of less than a day (e.g. styrene, C-8 and C-9 aromatics) feature prominently in several source profiles indicating that most of the transport sector emission were less than a day old when they reached the receptor site. Other compounds with

longer lifetimes such as toluene (2 days), benzene (6 days) or acetonitrile (months) can reach the site from more distant sources. The wheat residue burning source shows the highest cross correlation with the regional fire counts for a lag time of 2 days indicating that emissions from distant sources can and do impact the site with a time lag. The chosen fetch region includes the areas where the maximum number of wheat residue burning fire counts are observed while avoiding a size that is too large to be consistent with the relatively unaltered signature of some of the other PMF source profiles.

Annual emissions were available for EDGAR (2012) and GAINS (2010), whereas, REAS provided monthly data (May 2008). However, Figure S6 shows that despite providing monthly data, the REAS emission inventory has very little seasonality for any of the sources.

To facilitate the comparison of the PMF output of the month of May which is affected by a strongly seasonal source (crop residue burning) with emission inventories that provide only annual data as of now, we calculate hypothetical pie charts which attribute annual crop residue burning emissions over the region only to the 2.5 months when crop residue burning actually occurs (middle of October to end of November and May).

## 3 Results and Discussion

### 3.1 Split up of VOC Emission Sources in Mohali and their contribution to Ozone and SOA Formation Potential

Figure 4 (a) shows the percent contribution of the different sectors to ambient VOC mass concentration loadings during May 2012 in Mohali, while Figure S7 shows a time series of the total VOC mass contributed by the individual factors to the overall mass. The two traffic factors combined together were found to be the strongest contributors to the total VOC mass concentration (25.1 %) followed by biofuel use and waste disposal factor (23.2 %), wheat-residue burning (22.4 %), the mixed daytime factor (15.7 %) and industrial emissions (11.8 %), with the residual not apportioned VOC mass only amounting to 1.7 % of the total. Early source receptor modelling studies from India attributed a slightly larger share 26-58 % of the total VOC mass to traffic related emissions (Srivastava, 2004; Srivastava et al., 2005), suggesting that the progression to the emission norms Bharat stage III & IV (which are equivalent to Euro 3 and Euro 4 norms, http://cpcb.nic.in/vehicular-exhaust/) may have brought down VOC emissions from the traffic sector.

Figure 4 (b) shows the contribution of the different sectors to the ozone formation potential during May 2012 in Mohali. Wheat residue burning factor was found to be the largest contributor to the ozone formation potential (32.4 %) and has been shown to enhance ambient tropospheric ozone mixing ratios by 19 ppb (Kumar et al., 2016). Both traffic

sources combined, the mixed daytime sources, biofuel use & waste disposal, and industrial emissions and solvent use contributed 21.9 %, 20.3 %, 18.1 % and 7.3 %, respectively, to the ozone formation potential. It is clear that in order to bring ozone levels into compliance with the NAAQS, the wheat residue burning source of ozone precursors deserves the largest attention at this point, but the transport sector and biofuel use and waste disposal should not be neglected, either.

Figure 4 (c) shows the contribution of the different sectors to the SOA formation potential ($\sim 32\,\mu g\,m^{-3}$) under low $NO_x$ conditions. Traffic is the single largest contributor and is responsible for contributing 59.0 % of the SOA formation potential followed by biofuel use and waste disposal (14.9 %), wheat residue burning (13.9 %), industrial emissions and solvent use (10.1 %) and the mixed daytime factor (2.2 %). While the calculated SOA formation potential particularly from transport sector emissions (Ensberg et al., 2014) and aromatic compounds (Li et al., 2017; Li and Cocker III, 2018) is affected by large uncertainties and may depend in a non-linear fashion on NOx and VOC concentrations (Xu et al., 2015) our calculated SOA formation potential seem to indicate that SOA formation could contribute significantly to the average $PM_{2.5}$ mass loading (104 $\mu g\,m^{-3}$).

## 3.2    Factor 1 - Biofuel use & waste disposal

The biofuel use and waste disposal factor combines two sources with similar source profiles and high spatio-temporal overlap into one factor. As discussed previously for other South Asian atmospheric environments (Sarkar et al., 2017), the source contributions of domestic biofuel use and domestic waste burning are difficult to segregate. Figure 5 shows a weak bimodal behaviour with an early morning and late evening peak for this factor, as both domestic biofuel use and waste disposal fires peak in the early morning and in the evening hours (Nagpure et al., 2015). The highest conditional probability for this factor is from the North (>0.4), the direction of the Dadu Majra landfill in Chandigarh, followed by the wind direction NW where a large village (Mauli Baidwan) can be found within 1  km of the receptor and NE, the direction of Panchkula's garbage dump in Sector 23. This and the fact that the average contribution of this factor remains above 56 $\mu g\,m^{-3}$ throughout the night indicates that garbage burning contributes significantly to the biofuel use & waste disposal factor.

Figure 3 and Figure 6 show that this factor explains a significant share of the mass of acetonitrile (a biomass burning tracer), aldehydes, ketones, acids, propyne and propene in the PMF model. For propene (60%), aldehydes (85%) and ketones (68%) the residential sector is the dominant source in the most recent speciated emission inventory EDGARv4.3.2. The percentage share for aldehydes and ketones in the inventory is higher than its share in the PMF because the agricultural residue burning source of these compounds is currently missing in the inventory. For acids, however, the residential fuel usage source in the inventory (0.5%) is dwarfed by solvent use associated emissions (96%), while in the PMF the two biomass burning sources (residential biofuel use and waste disposal and wheat residue burning) account for almost 69 % of the total acids in the model. High emission of oxygenated VOCs have been reported previously for source profiles of biofuel-stoves (Wang et al., 2009; Paulot et al., 2011; Stockwell et al., 2016) open waste burning (Sharma et al., 2019) and PMF factors' results of residential biofuel use and waste disposal factor in Kathmandu, Nepal (Sarkar et al., 2017).

It should be noted, that this factor is responsible for approximately 25 % of the total benzene emissions in our PMF model, while emission inventories attribute a larger share (39-74%) of this compound to this source. Since benzene is an identified Group-1 carcinogen (IARC, 1987) and emissions occur within the household itself (domestic cooking) or within close proximity of the house (waste disposal) this factor deserves special attention in programs targeted at emission reductions. However, the impact of such emission reductions in the residential and waste management sector on human benzene exposure are likely to be overestimated by modelling studies using present day emission inventories, as the inventories attribute 39-74% of the benzene emissions to residential fuel usage and waste disposal, while the PMF suggests the transport sector is the largest benzene source (Figure S8a). Direct emission of isocyanic acid, a highly toxic emerging contaminant and its photochemical precursors (Alkyl amines and Amides) was observed from this source and explained 18 % of the isocyanic acid mass concentration and 7-15 % of all the alkyl amines and amides in the PMF model, respectively.

## 3.3    Factor 2 - Wheat Residue burning

Wheat residue burning takes place every year in the NW-IGP in the post-harvest season and generally peaks in the month of May. It has been shown that wheat residue burning has a major impact on both ozone mixing ratios (Kumar et al., 2016) and VOC mixing ratios and hydroxyl radical reactivity (Kumar et al., 2018) and results in a large suite of unknown ($\sim 40$ %) and poorly quantified reactive gaseous emissions. Wheat residue burning emissions are transported to the receptor site from a large fetch region and often with a significant lag time. Hence, there is no strong conditional probability for enhancements from any specific wind direction (Figure 5).

Figure 3 and Figure 6 show that the wheat residue burning factor explains a significant share of all acids, amines/amides, several ketones, and aldehydes, isoprene/furan, monoterpenes, acetonitrile, propene, styrene and phenol in the PMF model. This makes wheat residue burning the largest contributor to the human exposure to isocyanic acid in the month of May both through direct

emissions of isocyanic acid and by virtue of being the largest source for its photochemical precursors.

In the EDGARv4.3.2 the agricultural residue burning source of ketones,aldehydes and acids is missing. On the other hand agricultural waste burning appears to be the dominant anthropogenic isoprene source (94 %) in the EDGARv4.3.2 inventory while in our PMF model residential biofuel usage and the transport sector are equally important contributors to the isoprene/furan mass. The monoterpene emissions from agricultural residue burning (6 %) in the EDGARv4.3.2 inventory are dwarfed by emissions from solvent use (90 %), while in our PMF solution wheat residue burning and the transport sector appear to be the dominant anthropogenic sources of signals at m/z 81 and 137.

## 3.4 Factor 3 - Industrial emissions and solvent use

The source fingerprint of the industrial emissions and solvent use factor is dominated by methanol (7.3 $\mu g \, m^{-3}$), acetic acid (3.9 $\mu g \, m^{-3}$) and acetone (2.9 $\mu g \, m^{-3}$). This points towards solvent use (Gaimoz et al., 2011) and/or polymer manufacturing (Sarkar et al., 2017) contributing to the industrial emission and solvent use factor. In addition, Figure 3 and Figure 6 show that this factor explains a significant fraction of the benzene (20 %) and acetonitrile (17 %) mass in the PMF model. While both are known for their use as solvents (Brown et al., 2007), they can also be emitted from the combustion. The EDGARv4.3.2 emission inventory has a strong industrial and solvent source of toluene, xylenes, acids, formaldehyde and monoterpenes which is not reflected with equal strength in our PMF solution.

The correlation of the industrial emissions and solvent use factor with the $SO_2$ time series (R= 0.6), indicates that the emissions of coal or biofuel burning in industrial units and/or coal fired power plants may also be contributing to this factor profile. Figure 5 shows that the highest conditional probability of this factor is to the South East direction (120 ° -150 ° wind sector). The receptor site is downwind of a 600 MW coal fired power plant located in Jagadhri (80 km SE) as well as downwind of several industrial areas and brick kiln clusters located around Dera Bassi (15 km), Lalru (20 km) and Jagadhari (80 km) when the wind blows from this direction. In the Kathmandu valley, biofuel co-fired brick kilns explained a significant fraction of the benzene and acetonitrile mass (Sarkar et al., 2017) and the factor profile shows a moderate correlation with the source signature of brick kiln emissions (R=0.5), hence a combustion contribution from brick kilns to the factor profile cannot be ruled out. The diel profile broadly reflects boundary layer dynamics with factor contributions increasing continuously throughout the night indicating a buildup of constant emissions in the nocturnal boundary layer. Factor contributions peak in the early morning (32-49 $\mu g \, m^{-3}$ between 5-9 am local time) and the factor contribution of this factor decreases from 9 am onwards after the breakup of the nocturnal boundary layer. This factor has

higher average than the median factor contributions at night due to strong plumes ($\sim 375 \, \mu g \, m^{-3}$) reaching the receptor when it is downwind of the industrial sector but not during other nights when the wind direction is from rural Punjab (NW) or the urban sector (NE).

## 3.5 Factor 4 and 5 - cars and two-wheelers

The factor profile of the 4-wheeler factor explains a significant share of all aromatic compounds in the PMF model. The factor represents a mixture of multiple components contributed by fuel exhaust and fuel evaporative running losses from vehicles and resembles ambient air samples from a busy traffic intersection. Similar profiles have been observed during field measurements in Beirut, Lebanon (Salameh et al., 2014, 2016) and Hong Kong (Ho et al., 2004). The highest conditional probability (Figure7) is observed for the Chandigarh wind sector (0-90 °). As reported previously from Mexico City during the Milagro campaign (Bon et al., 2011), a significant mass of methanol (4.3 $\mu g \, m^{-3}$) and other oxygenated VOCs were present in the traffic emissions factor. The fact that this factor explains 28 % of the total m/z 57 is consistent with the gasoline additive MTBE being detected at this m/z ratio as an interference to acrolein/methylketone (Karl et al., 2003; Warneke et al., 2003, 2005; Rogers et al., 2006). Signals at m/z 31, 47, 59, 61, 73, 87 in aged traffic plumes can be attributed to formaldehyde, formic acid, glyoxal, acetic acid, methylglyoxal and 2-butanedione which are products of the gas phase oxidation of toluene, C-8 and C-9 aromatic compounds (Bethel et al., 2000; Ervens et al., 2004). In addition, car exhaust also explained 34 % of the propyne mass in the model.

Factor 5, 2-wheeler exhaust, explains 50 % of the total toluene mass as well as 17 %, 12 % and 9 %, of the total C-8 aromatics, benzene and C-9 aromatics in the PMF model, respectively. The factor shows a signal at m/z 61 (acetic acid) which may partially be due to fragmentation of octane or ethyl acetate (Warneke et al., 2003; Rogers et al., 2006) which could be present in fuel. The mass has also been attributed to acetic acid in a previous study of diesel tailpipe emissions (Jobson et al., 2005). Nevertheless, it still seems that the 2-wheeler factor profile has a higher contribution from oxidised compounds compared to the car factor profile indicating that the plumes are typically more aged. Figure 7 shows that this factor displays higher conditional probability than the car factor towards the towns Kharar (8 km N), Dera Bassi (15 km SE) and Lalru (20 km SE), and a lower conditional probability than the car factor towards Chandigarh (NE) indicating 2-wheelers are more abundant in small towns, while cars dominate the traffic emissions in urban Chandigarh.

Figure 7 illustrates that both the traffic factors show bimodal peaks in morning (19 $\mu g \, m^{-3}$ at 5-9 am local time) and evening (38 $\mu g \, m^{-3}$ at 7-9 pm local time) during peak traffic hours. Mass loadings during evening rush hour are

higher than during morning rush hour, because peak morning traffic occurs after the breakup of the nocturnal boundary layer, while in the evening emissions accumulate in the shallow nocturnal boundary layer. When the wind blows from the urban sector (0-90 °) during peak traffic hour (7-9 pm) peak factor contributions of >260 $\mu g\,m^{-3}$ for cars and >150 $\mu g\,m^{-3}$ for 2-wheelers are observed.

As can be seen from Figure 6, the two traffic factors jointly explain 47 %, 80 %, 70 % and 67 % of the total benzene, toluene, C-8 and C-9 aromatic compounds in the model consistent with findings from the Kathmandu valley that traffic, not residential biofuel use and waste disposal is the more important source of aromatic compounds in South Asia. It is also clear that despite stringent regulations, the transport sector in the region is still the largest contributor to human benzene exposure. It can be seen from Figure S8a-d that at present, various emission inventories consider the transport sector to be a minor source of benzene (10-16%). The EDGAR v4.3.2 emission inventory also considers the transport sector to be only a minor source of, toluene (11-15%) and xylenes (17-22%). Residential fuel usage, industries and solvent use are considered to be the most significant year around source of benzene, toluene and xylenes in Edgar v4.3.2. Agricultural residue burning becomes the most significant source of all aromatic compounds in the EDGAR v4.3.2 emission inventory when crop residue burning emissions are treated as occurring during crop residue burning season only, which may imply that the annual emissions of aromatic compounds from the stubble burning may be overestimated. REAS v.2.1 appears to be overestimating the residential fuel burning contribution to benzene and toluene emissions and the solvent usage contribution to toluene emissions. However, it captures the contribution of the transport sector to xylenes and trimethylbenzenes emissions well.

## 3.6    Factor 6 - mixed daytime sources

Figures 4 and 6 show that mixed daytime sources comprising of biogenic emissions and photochemically formed compounds explained 22 % of the monoterpenes and 25 % of the measured isoprene, respectively. Isoprene has a short chemical lifetime of 1.5 hours during the day and 16 % and 11 % of its first generation oxidation products MVK and MEK (Kesselmeier and Staudt, 1999) were also attributed to this factor . In addition, the mixed daytime factor explains 41 %, 44 %, 24 % and 22 % of the total formaldehyde, formic acid/ethanol, methanol and acetone mass, respectively. Photochemically formed isocyanic acid, formamide, acetamide and propanamide explain a slightly lower fraction (27-37 %) of the total mass concentration of these compounds compared to what has been reported from wintertime Kathmandu valley (36-41 %). Figure 7 illustrates that the mixed daytime factor peaks between 9 am and 4 pm and shows a slightly enhanced conditional probability for the 180 ° -330 ° rural wind

sector (0.2-0.3) due to agroforestry plantations of poplar in the rural landscape.

## 3.7    Comparison of PMF source factors with existing Emission Inventories

Figure 8 shows pie charts depicting the contribution of different sectors to the total VOC mass burden for the emission inventories and our PMF output. Biofuel use and waste disposal were responsible for 28.1 % of the mass in our PMF but 39 %, 44.2 % and 41.7 % of the mass in EDGARv4.3.2, GAINS and REASv2.1 respectively. The contribution of crop residue burning (27.1 %) to the VOC mass in the month of May would be highly underestimated by both GAINS (7 %) and EDGARv4.3.2 (6 %) if the annual emissions are attributed equally to all months of the year. However, if both emission inventories would attribute their annual crop residue burning emissions over the region only to the 2.5 months when crop residue burning actually occurs (middle of October to end of November and May), these emission inventories could be reconciled with the PMF solution, as emissions in May would amount to 26.5 % and 23 % of the monthly VOC emissions for the month of May for GAINS and EDGARv4.3.2, respectively as shown in Figure 8. At the same time the percentage share of domestic fuel use and waste disposal would drop to 32 % and 35 % in EDGARv4.3.2 and GAINS, respectively and the contribution of industrial emissions and solvent use would drop to 18 % in GAINS and 30 % in EDGAR, respectively. Our PMF (14.3 %) solution indicates that industrial emissions and solvent usage (14.3%) are currently overestimated in all emission inventories but are closest to GAINS (540 $Gg\,y^{-1}$, 18%) for industrial emissions and solvent use . For domestic biofuel use and waste disposal EDGARv4.3.2 (968 $Gg\,y^{-1}$, 32%) appears to agree best with our PMF solution. For wheat residue burning GAINS agrees well with our PMF output, while the agricultural waste burning emissions of some of the detected compound groups (ketones, aldehydes and acids) appear to be missing in the EDGARv4.3.2 inventory. Our PMF solution for road transport sector emissions (30.5 %) lies in between the estimates of GAINS (558 $Gg\,y^{-1}$, 24 %) and REAS (1230 $Gg\,y^{-1}$, 36.2 %), possibly, because not all pre-2000 super-emitters for which the 20-year vehicle lifetime has been exceeded have been retired as planned.

Overall it appears that none of the emission inventories is ideal at the present. Our PMF solution suggests that transport sector emissions may be are underestimated by GAINS and EDGARv4.3.2, while the combined effect of residential biofuel use and waste disposal emissions as well as the VOC burden associated with solvent use may be overestimated by all emission inventories. Similar results have been reported previously. Sarkar and co-workers (Sarkar et al., 2017) reported an underestimation of transport sector emissions for the REAS and EDGAR emission inventory for the Kathmandu valley in Nepal and an overestimation of the resi-

dential biofuel use and waste disposal source in all emission inventories, while Gaimoz and co-workers (Gaimoz et al., 2011) reported an overestimation of the VOC emissions from solvent use in Paris.

## 4  Conclusions

Our results highlight that for accurate air quality forecasting and modelling it is essential that emissions are attributed only to the months in which the activity actually occurs. This is important for emissions from crop residue burning (which occur in May and from Mid-October to the end of November). Annually averaged emissions are unlikely to yield accurate air quality forecast in regions affected by such seasonal events. At present, more specialized fire emission inventories such as FINN (Wiedinmyer et al., 2011) must be used to account for the full seasonality and day to day variations of open burning emissions. We also demonstrate, that the source profiles obtained as PMF output can be validated and matched against samples collected at the potential sources to validate the factor identification.

For the human class I carcinogen benzene, the traffic factor alone contributed to 47 % of the total benzene mass at this receptor site followed by residential biofuel use and waste disposal (25 %) and industrial emissions and solvent use (20 %). This stands in stark contrast to various emission inventories which estimate the transport sector contribution to the benzene exposure as ( 10%) and consider residential biofuel use, agricultural residue burning and industries to be more important benzene sources. Since the annual NAAQS for benzene is exceeded at this receptor site (Chandra and Sinha, 2016), all three sectors must be targeted for emission reductions.

For the emerging contaminant isocyanic acid, photochemical formation from precursors (37 %), wheat residue burning (25 %) and biofuel usage and waste disposal (18 %) were the largest contributors to human exposure. The monthly average isocyanic mixing ratio of 1.4 ppb exceeds concentrations that can, after dissociation at blood pH, result in blood cyanate ion concentrations (Roberts et al., 2011) high enough to produce significant health effects in humans (Wang et al., 2007) such as atherosclerosis, cataracts and rheumatoid arthritis due to protein damage. Peak mixing ratios of this compound exceed 3 ppb in some night time wheat residue burning plumes. Wheat residue burning was also the single largest source of the photochemical precursors of isocyanic acid, namely, formamide, acetamide and propanamide, indicating that this source must be most urgently targeted to reduce human concentration exposure to isocyanic acid.

Overall it appears that none of the emission inventories is ideal at the present. Our PMF solution suggests that transport sector emissions may be underestimated by GAINSv5.0 and EDGARv4.3.2, while the combined effect of residential biofuel use and waste disposal emissions as well as the VOC burden associated with solvent use may be overestimated by all emission inventories. Agricultural waste burning emissions of some of the detected compound groups (ketones, aldehydes and acids) are currently missing in the EDGARv4.3.2 inventory while aromatic emissions from the same source appear to be overestimated. Thus, large improvements are required in existing emission inventories for correct source attribution and inclusion of missing compounds over this densely populated region of the world.

*Data availability.*  Data is available from the corresponding author upon request.

*Author contributions.*  Pallavi performed the analysis and wrote the first draft of the paper. Dr. Baerbel Sinha conceived the analysis and revised the paper draft. Dr. Vinayak Sinha collected the data and commented on the paper draft.

*Competing interests.*  The authors have no competing interests to declare.

*Acknowledgements.*  We acknowledge the IISER Mohali Atmospheric Chemistry facility for data and the Ministry of Human Resource Development (MHRD), India for funding the facility. Pallavi acknowledges IISER Mohali for Institute PhD fellowship. This work was also partially supported through grant (SPLICE) DST/CCP/MRDP/100/2017(G) under the National Mission on Strategic knowledge for Climate Change (NMSKCC) MRDP Program of the Department of Science and Technology, India.

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

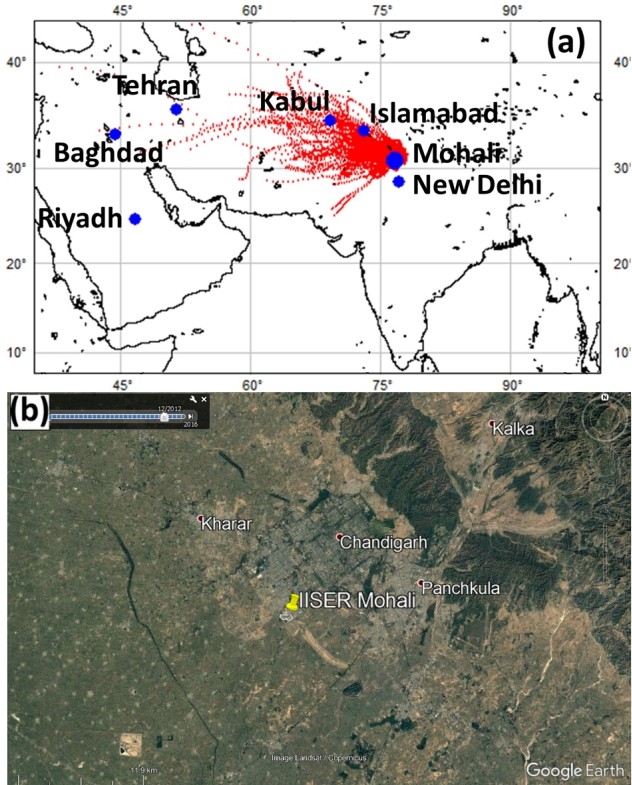

**Figure 1.** (a) Mohali located on Indian Subcontinent with the overlaid 72 h air mass back trajectories for May 2012 at 09:00 LT and 23:00 LT (UTC+5:30) (b) Precise location of IISER-Mohali Atmospheric chemistry facility (30.667 °N, 76.729 °E, 310 m above mean sea level) with nearby cities on Google Earth imagery. The campus of IISER Mohali is outlined in white.

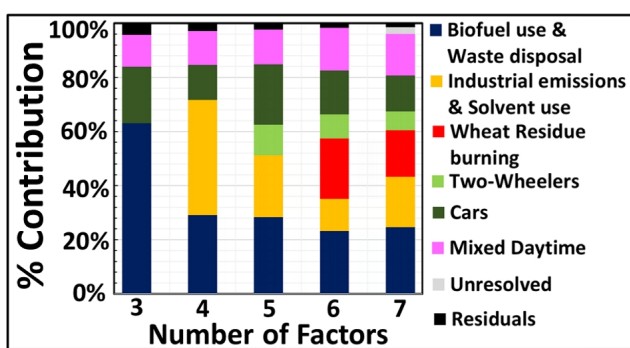

**Figure 2.** Percentage contribution assignment for various PMF factor number solutions (3-7) to the corresponding VOC emission sources.

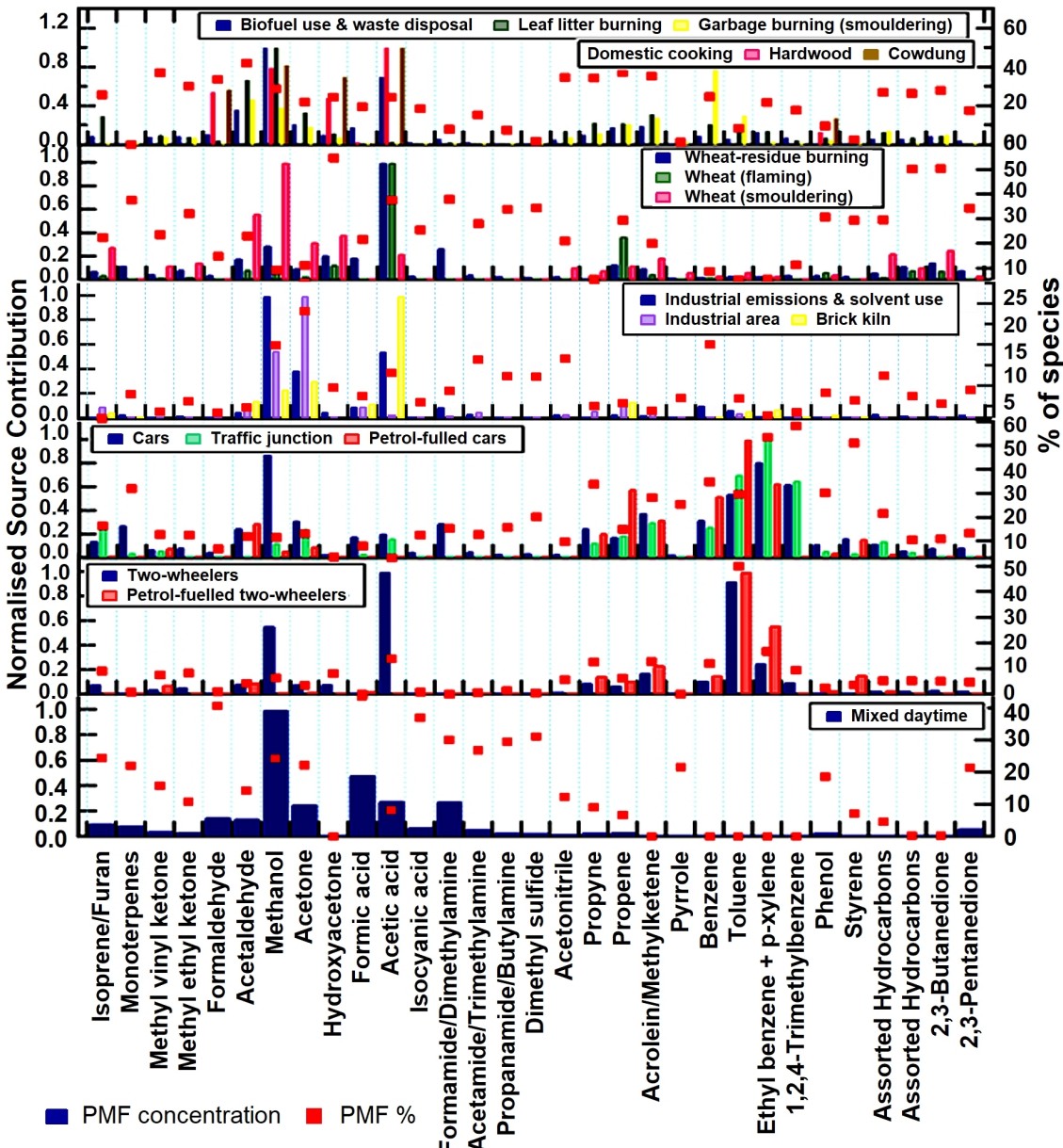

**Figure 3.** Factor profile composition for (6) PMF resolved factors at IISER-Mohali. It displays the normalized source fingerprints of the PMF factors (dark blue) and samples collected at source (in various colours) in bar-chart form. The value of the normalized species contribution is depicted on the left hand axis. The percentage of each species explained by each of the PMF factors is displayed in the form of a red square to be read from the right hand axis.

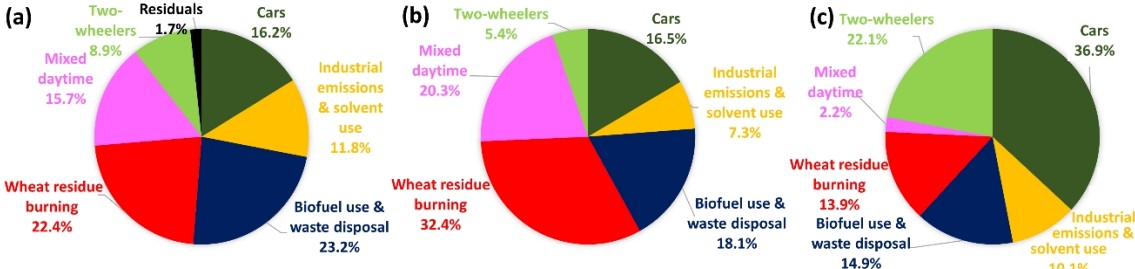

**Figure 4.** (a) Source contribution to the ambient VOC loading at the receptor site. (b) Ozone formation potential for PMF derived sources (c) SOA potential for PMF factors.

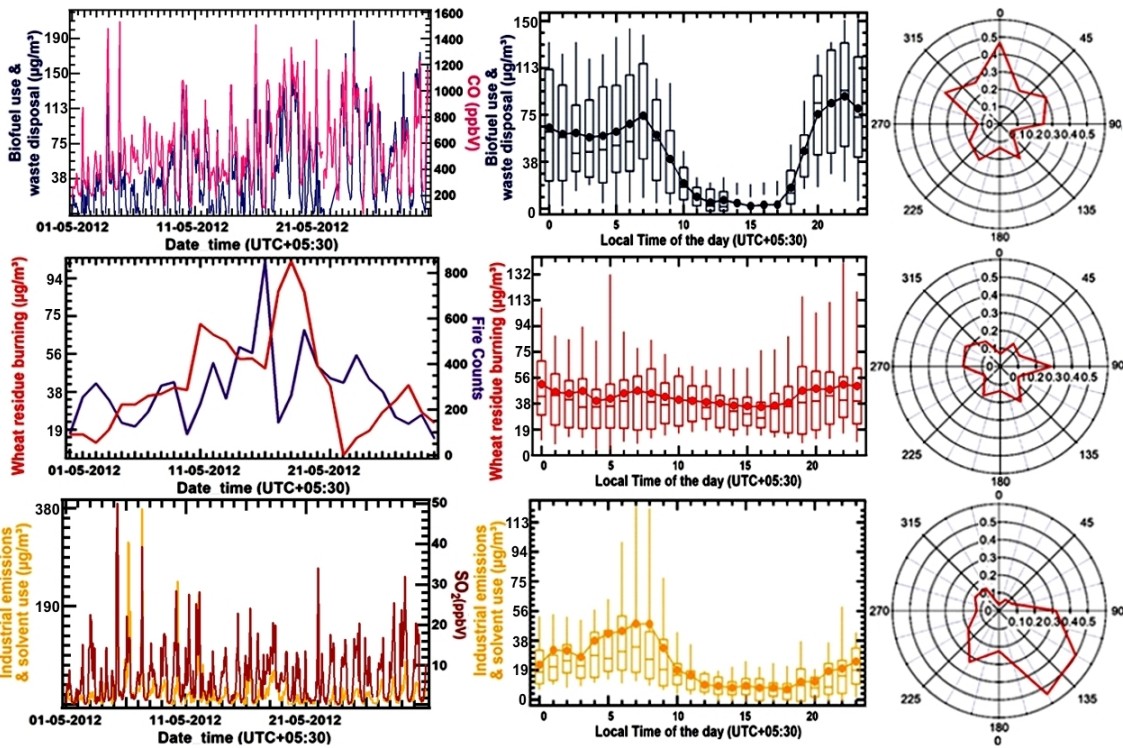

**Figure 5.** Factor contribution time series, factor diel variability and CPF plot for PMF Factor 1 (Biofuel use and waste disposal), PMF Factor2 (Wheat-residue burning) and PMF Factor3 (Industrial emissions and solvent use) for May2012. The time series of PMF factor's hourly mass in $\mu g\,m^{-3}$ is plotted against independent tracer species CO (in ppbv) for the biofuel use and waste disposal factor, daily fire counts for the wheat residue burning factor and $SO_2$ (in ppbv) for the industrial emission and solvent use factor. The Diel box and whisker plot shows the statistical parameters of factor's hourly mass contribution in $\mu g\,m^{-3}$ for every hour of the day plotted against the start time of the hour. The width of the box gives 25th and 75th percentiles, 50th percentile partitions the box; whiskers represent 10th and 90th percentiles of the dataset and average values are given by solid circles.

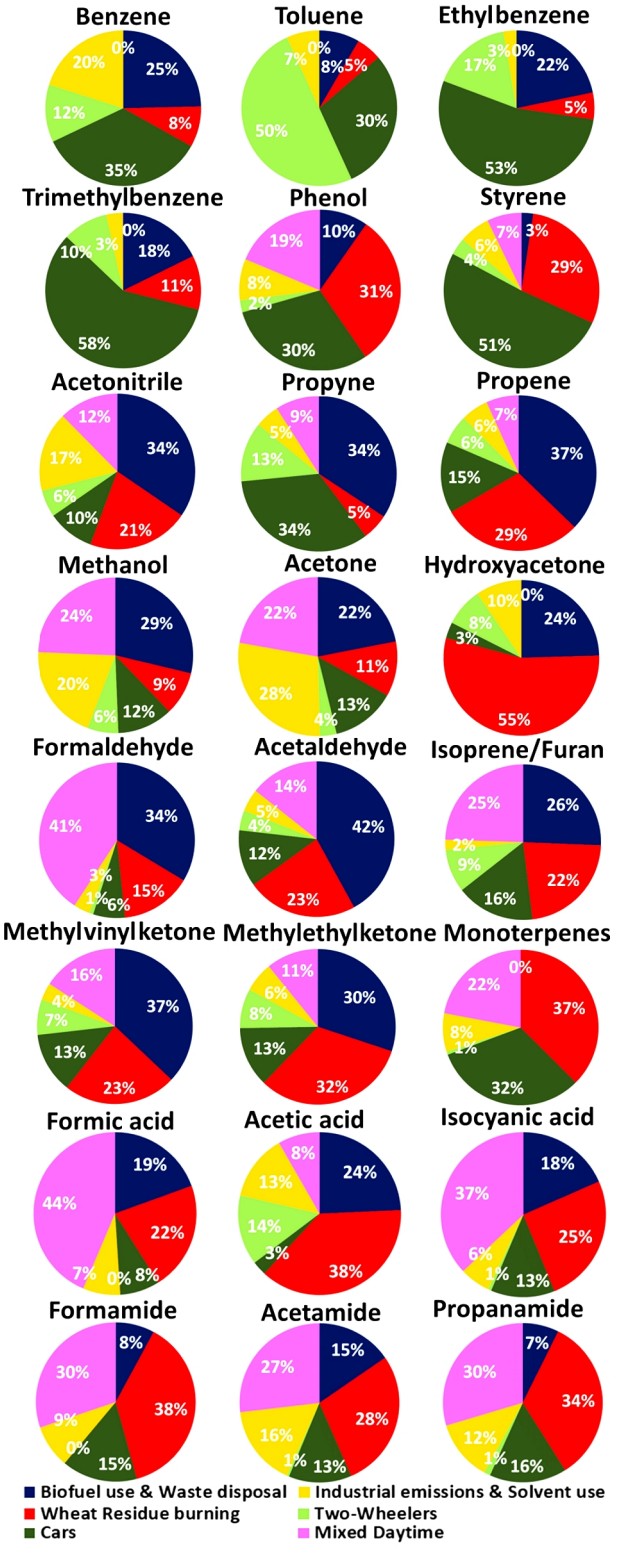

**Figure 6.** Contribution of individual PMF derived source factors to the total mass of different VOCs.

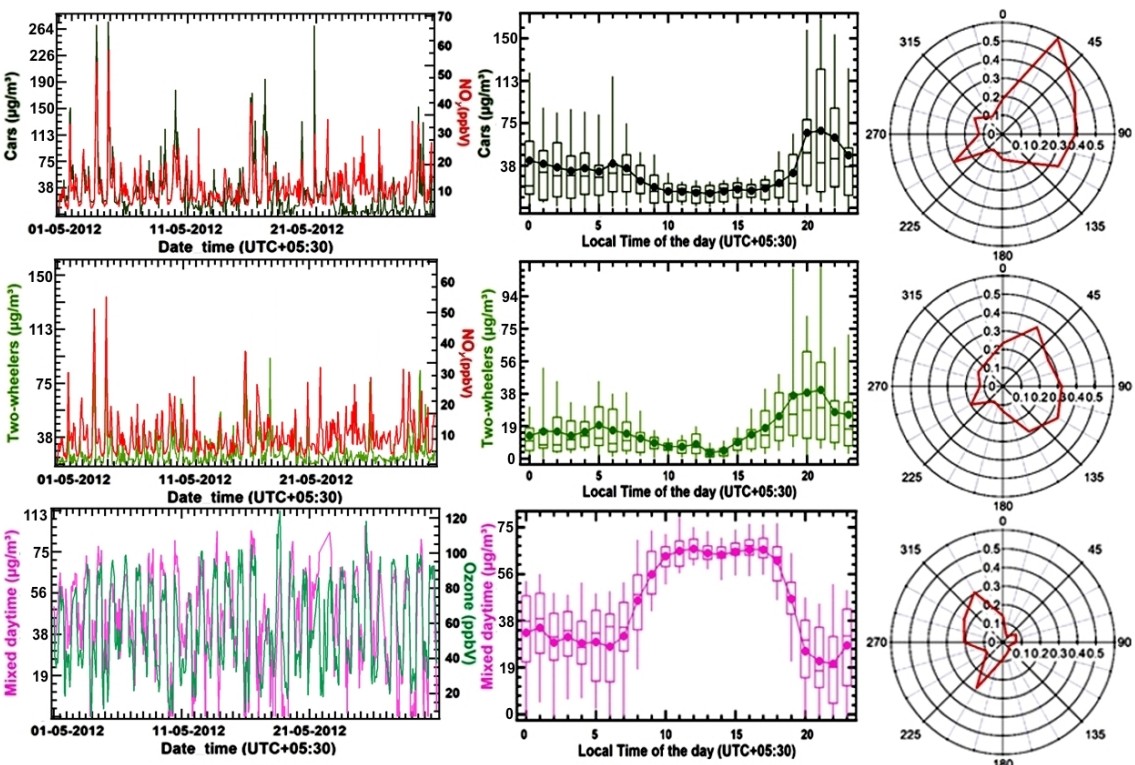

**Figure 7.** Factor contribution time series, factor diel variability and CPF plot for PMF Factor 4 and Factor 5 (Cars and two-wheelers) and PMF Factor 6 (Mixed daytime) for May2012. The time series of PMF factor's hourly mass in $\mu g\,m^{-3}$ is plotted against independent tracer species $NO_y$ (in ppbv) for the car and two-wheeler factor and and $O_3$ (in ppbv) for the mixed daytime factor. The Diel box and whisker plot shows the statistical parameters of factor's hourly mass contribution in $\mu g\,m^{-3}$ for every hour of the day plotted against the start time of the hour. The width of the box gives 25th and 75th percentiles, 50th percentile partitions the box; whiskers represent 10th and 90th percentiles of the dataset and average values are given by solid circles.

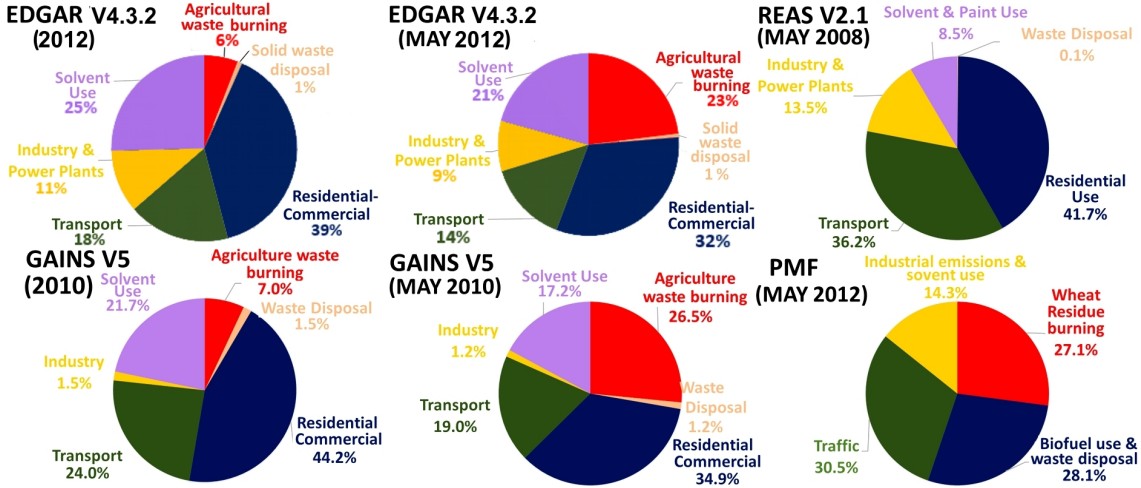

**Figure 8.** Comparison of PMF derived VOC source contribution to the EDGAR, REAS and GAINS Emission Inventory Database.