# Peer review of "Source apportionment of volatile organic compounds in the north-west Indo-Gangetic Plain using positive matrix factorisation model"

_Atmospheric Chemistry and Physics, 2019_

## Referee Comment (RC1) · Anonymous Referee #1 · 24 Jul 2019

**1   Overall comment**

This study focuses on the source apportionment of VOCs measurements at a suburban site in the North-West Indo–Gangetic Plain. The period studied is the month of May 2012. Authors use a Positive Matrix Factorization Model (PMF) to resolve source contributions to VOCs, perform a conditional probability functional analysis to locate the different sources and calculate the ozone and secondary organic aerosols formation potential. Moreover, results of PMF are compared with the source apportionment of three different emission inventory estimates.

Overall, the analysis performed is interesting and valuable. However, the manuscript

needs improvements in the logical framing of the work with respect to its contribution and implications to the field. Also the introduction and the results need to be improved in this sense. I recommend publication after the authors have addressed the following substantive concerns/comments on their manuscript.

**2 Major comments**

1. ABSTRACT - the abstract is a bit too technical. I recommend to focus more on the big picture and major findings and implication of the paper (as outlined in the conclusions).

2. INTRODUCTION - the introduction should better frame the background of the study, its motivation and what is the new contribution of the work. In particular:

   - Only one source receptor modelling study that has been cited is in the region of the study (Srivastava et al., 2005) . Are there any source receptor modelling or more general studies that focus on VOCs over the IGP? If yes, they should be acknowledged. If no, this should be underlined.
   - VOCs source apportionment estimates for the region under study are presented for different emissions inventories. However it is claimed 'Considering the large discrepancies between bottom up inventories and different source receptor modelling studies', when 2/3 source receptor models studies presented so far are out of the understudied region. This claim need to be justified, or more appropriate studies need to be cited.
   - The study take into consideration a specific month, May 2012. It is needed to explain why this month is important for the region under study and which general conclusions can be made from it.

- The aims of the paper need to be better outlined (e.g. in the last paragraph of the introduction the comparison with emissions inventories is not mentioned in the objectives).

3. METHODS - The description of methods should be revised in its content. In particular:

   - Why have authors chosen to use the US EPA PMF 5.0 model? A brief motivation and description of the model need to be provided along with relevant references.

   - Almost the entire part of the methodology in Section 2.4 and 2.5 is left to the supplement or to other studies. Since it is a fundamental part of the methodology used in this the study, I would suggest to expand these sections. On the other hand, the detailed description in section 2.2. is not really relevant for this study, and should be cut/shortened.

   - The description of the methodology used to compare results of this study with the emission inventories estimates should be outlined.

4. RESULTS AND DISCUSSIONS -

   - Overall results are too descriptive, and there are repetitions of information that figures already provide. I suggest to focus more on what can be deduced from the analysis rather then on its description.

   - Section 3.8 presents the comparison between the source apportionment study and emission inventories estimates, i.e. a point vs gridded data. Is it sufficient to filter gridded data for LAT LONG from which air mass trajectories reach the site within one day to make the comparison reliable? Moreover, the study consider May 2012, while emissions inventory data are for 2008/2010. Which are the uncertainties in using these approaches in the comparison? Authors should justify and better describe these choices.

5. CONCLUSIONS - It would be more valuable for the reader if the authors focused more on the achievements and implications of the results. The last paragraph of 3.8 may be included in the conclusions rather then in results.

**3   Minor comments**

1. First author name (Pallavi) is missing.

2. Page 2 line 5 '...deserve further study' this sentence need citation.

3. Page 2 line 31 '...and strong photochemistry' this sentence need citation.

4. Section 2.3: need to add cross references to Table S3, Figure S4 a, b c.

5. Pag. 9 line 16 'However, Figure S5..'. It is Figure S6 in the Supplement.

6. Figure 1 (b): add lat - long grid. It may be worth to add in the caption the exact coordinates of the site.

---

## Referee Comment (RC2) · Anonymous Referee #2 · 9 Aug 2019

The article titled "Source apportionment of volatile organic compounds in the north-west Indo–Gangetic Plain using positive matrix factorisation model" by Pallavi et al., is generally well written and contains some useful information. VOC source apportionment studies are sparse in India, and this study could encourage more such studies in future which is required to understand the VOCs impact on air quality. However, on many occasions, authors seem to over-interpret the results and have drawn some rather farfetched conclusions. I would recommend publication provided my concerns are being addressed satisfactorily.

Abstract:

[Figure]

Numbers can be presented in a better way for ease of reading. Authors can put the percentage contribution of different factors/parameters in the parenthesis beside them.

Methods:

Sec 2.3: Line 20, why 20%? Please explain and incorporate in the manuscript as well.

Line 20, more than 50% of the measured species (18 of 32) are weak, isn't that going to influence the robustness & reliability of the PMF output?

Line 24, Why the authors chose to remove missing values instead of replacing them with some other values as mentioned in the literature? Is this the standard practice? what is the % of missing values in the total sampled points?

Sec 3.3

Line 1-3, R= 0.4 is not a good correlation, at best it can be termed as moderate. Please rewrite the explanation on why fire count is the best tracer for factor 2.

Sec 3.7

Line 7, in PM2.5, 2.5 should be subscript

Line 7, I don't think the way SOA being calculated enable the authors to make such strong quantitative assertion about the SOA contribution to PM2.5 in Mohali. At best, the adopted method can provide a qualitative and comparative assessment of SOA production efficiency among different PMF factors. I would suggest to remove or modify line 6-8 to reflect this.

Sec 3.8

I am not sure about the utility or purpose of this section. Are authors trying to use this comparison as another tool for PMF results validation? Or to suggest which inventory is better? Every emission inventory is developed based on some underlying assumptions and approximations. I would rather be very surprised if a single site based study can

reproduce or match the emission inventory values. It is quite expected that differences will be there and even a perfect match doesn't necessarily validate emission inventories or the PMF results, especially in a complex source environment as in India. Several assumptive statements were made to explain the mismatch/less match between PMF results and emission inventory values. So, based on this comparison one can't really assert which inventory is better or more representative than others. Authors should remove or rephrase the section to reflect those concerns.

Figures:

I want to see Q/Qexp plot in SI.

Fig. 7: Why the evening peaks in Car & Two-wheeler contributions are significantly more pronounced than morning hours?

---

## Author Comment (AC3) · 20 Sep 2019

The comment was uploaded in the form of a supplement:
https://www.atmos-chem-phys-discuss.net/acp-2019-343/acp-2019-343-AC3-supplement.pdf

---

## Author Response (AR1)

**Final Author response**

**Response to Anonymous Referee #1**

**Reviewer comment:**
**1 Overall comment**
This study focuses on the source apportionment of VOCs measurements at a suburban site in the North-West Indo-Gangetic Plain. The period studied is the month of May 2012. Authors use a Positive Matrix Factorization Model (PMF) to resolve source contributions to VOCs, perform a conditional probability functional analysis to locate the different sources and calculate the ozone and secondary organic aerosols formation potential. Moreover, results of PMF are compared with the source apportionment of three different emission inventory estimates.

Overall, the analysis performed is interesting and valuable. However, the manuscript needs improvements in the logical framing of the work with respect to its contribution and implications to the field. Also the introduction and the results need to be improved in this sense. I recommend publication after the authors have addressed the following substantive concerns/comments on their manuscript.

**Author response:** We sincerely thank the reviewer for his encouragement and the in-depth comments and suggestions which have greatly improved the clarity of the manuscript and have helped us to emphasize the implications of this study to the field more clearly.
The detailed response to each comment and changes made in the manuscript are listed below.

**2 Major comments**
1. ABSTRACT - the abstract is a bit too technical. I recommend to focus more on the big picture and major findings and implication of the paper (as outlined in the conclusions).
**Author response:** We appreciate this feedback and have revised the abstract in accordance with it. Revised abstract reads as follows:

**Changes in the manuscript:**
"In this study we undertook quantitative source apportionment for 32 volatile organic compounds (VOCs) measured at a suburban site in the densely populated North-West Indo-Gangetic Plain using the US EPA PMF 5.0 Model. Six sources were resolved by the PMF model. In descending order of their contribution to the total VOC burden these are "biofuel use and waste disposal" (23.2%), "wheat-residue burning" (22.4%), "cars" (16.2%), "mixed daytime sources" (15.7%), "industrial emissions and solvent use" (11.8%) and "two-wheelers" (8.6%).
Wheat residue burning is the largest contributor to the total ozone formation potential (26.2%). For the emerging contaminant isocyanic acid, photochemical formation from precursors (37%) and wheat residue burning (25%) were the largest contributors to human exposure. Wheat residue burning was also the single largest source of the photochemical precursors of isocyanic acid, namely, formamide, acetamide and propanamide, indicating that this source must be most urgently targeted to reduce human concentration exposure to isocyanic acid in the month of May. Our results highlight that for accurate air quality forecasting and modelling it is essential that emissions are attributed only to the months in

which the activity actually occurs. This is important both for emissions from crop residue burning (which occur in May and from Mid-October to the end of November).
The SOA formation potential is dominated by "cars" (36.9%) and "two-wheelers" (21.1%), which also jointly account for 47% of the human class I carcinogen benzene in the PMF model. This stands in stark contrast to various emission inventories which estimate the transport sector contribution to the benzene exposure as (~10%) and consider residential biofuel use, agricultural residue burning and industries to be more important benzene sources. Overall it appears that none of the emission inventories represent the regional emissions in an ideal manner. Our PMF solution suggests that transport sector emissions may be underestimated by GAINSv5.0 and EDGARv4.3.2 and overestimated by REASv2.1, while the combined effect of residential biofuel use and waste disposal emissions as well as the VOC burden associated with solvent use and industrial sources may be overestimated by all emission inventories. The agricultural waste burning emissions of some of the detected compound groups (ketones, aldehydes and acids) are missing in the EDGARv4.3.2 inventory."

**Reviewer comment:** 2. INTRODUCTION - the introduction should better frame the background of the study, its motivation and what is the new contribution of the work. In particular:
• Only one source receptor modelling study that has been cited is in the region of the study (Srivastava et al., 2005). Are there any source receptor modelling or more general studies that focus on VOCs over the IGP? If yes, they should be acknowledged. If no, this should be underlined.

**Author response:** We have cited all the source receptor modelling studies for VOC performed in India that are available in the peer reviewed literature. Beyond the studies we cited, there is one PMF study from the Kathmandu valley in Nepal (Sarkar et al. 2017) which represents a different environment and one attempted PMF study from the Eastern Himalayas (Sarkar et al. 2014, acpd-14-32133-2014), which did not make it into ACP. We have now made it more clear that no other VOC Source apportionment study in the IGP exists.
**Changes in the manuscript:** We inserted the following sentence after line 12 on page 2 "The only other source receptor modelling study in South Asia was conducted using a positive matrix factorisation model (EPA PMF5.0) with data collected in the Kathmandu valley, Nepal, as part of the SUSKAT campaign and attributed a negligible fraction of the anthropogenic VOC burden to residential biofuel usage (~14%). Instead different industrial sources including brick kilns (jointly 52%) and the transport sector (21%) were identified as the dominant VOC sources in the Kathmandu valley."

**Reviewer comment:** VOCs source apportionment estimates for the region under study are presented for different emissions inventories. However, it is claimed 'Considering the large discrepancies between bottom up inventories and different source receptor modelling studies', when 2/3 source receptor models studies presented so far are out of the understudied region. This claim need to be justified, or more appropriate studies need to be cited.
**Author response:** We agree with the reviewer that comparing performance of emissions inventories for the entire NW-IGP with a source receptor model study conducted in a specific megacity could introduce a bias. Since we have no other source receptor studies to fall back on, we have now made the comparison between the inventory and the source receptor modelling studies more specific and spatially accurate limiting it to the Delhi National Capital Region, Greater Mumbai and Greater Kolkata, when comparing with the source receptor

modelling studies of these cities, respectively and have revised the text from line 5-11 accordingly.

In response to a later comment by the same reviewer about the mismatch between the emission inventory year and the year of our study we have now substituted EDGAR v4.2 with the latest EDGAR v4.3.2 version (Huang et al. Atmos. Chem. Phys., 17, 7683–7701, 2017), with emissions for the year 2012, which appears to represent a significant improvement over the previous version. We have not included MIX Asia (Li et al. Atmos. Chem. Phys., 17, 935–963, 2017) since the NMVOC data of this inventory for India has been sourced from REASv2.1 without any changes. As a result of this update we have also revised the text of lines 11-17 in the introduction (not just the results and discussion section 3.8 and Figure 8).

**Changes in the manuscript:** The revised paragraphs now read:

"Previous source receptor modelling studies of VOC emission from India (Srivastava, 2004; Srivastava et al., 2005; Majumdar et al., 2009) produced results that conflicted strongly with the bottom up emission inventories, all of which contain significant emissions from residential fuel usage even when filtered for the New Delhi National Capital Region (41-45%), Greater Mumbai (32-36%) and Greater Kolkatta (33-59%). Transport sector emissions, according the bottom up emission inventories contribute only 15-35%, 17-43% and 6-14% to the total VOC emissions in New Delhi National Capital Region, Greater Mumbai and Greater Kolkatta, respectively. All previous studies from India employed a chemical mass balance (CMB) technique for ambient VOC source attribution and identified the transport sector as the main source of NMVOCs in the form of evaporative emissions (40-87%) in Mumbai (Srivastava, 2004), diesel internal combustion engines (26-58%) in Delhi (Srivastava et al., 2005) and roadway/refuelling exhaust (~40%) in Kolkata city (Majumdar et al., 2009). Except for the study performed in Kolkata which found a contribution of <10% from wood combustion, residential fuel usage was not identified as a potential VOC source in those source receptor modelling studies. The observed discrepancy could be partially caused by the fact a CMB is not necessarily an ideal tool for conducting source receptor modelling study in understudied environments as the model needs to be initialized with locally measured source profiles of all potentially significant sources. However, it is unlikely that this is the only reason for the discrepancies between source receptor modelling outcomes and emission inventories.

Different bottom up emission inventories also have large discrepancies between each other when extracted for the NW-IGP. For our study region (27.4-34.9 °N and 72-79.8 °E), EDGAR v4.3.2 estimates that the road transport sector contributes only 18% of the total anthropogenic VOC emissions (440 Gg $y^{-1}$), while REAS v2.1 (and MIX Asia) attribute 35.8% of the total anthropogenic VOC emissions (1230 Gg $y^{-1}$) to this sector. For industrial emissions and solvent use, GAINS has the lowest (540 Gg $y^{-1}$) and EDGAR v4.3.2 the highest absolute emissions (900 Gg $y^{-1}$). Crop residue burning as VOC source is missing in REAS but accounts for a 6% (145 Gg $y^{-1}$) and 7% (163 Gg $y^{-1}$) share of the annual VOC emissions in EDGAR and GAINS, respectively."

**Reviewer comment:** The study takes into consideration a specific month, May 2012. It is needed to explain why this month is important for the region under study and which general conclusions can be made from it.

**Author response:** The month of May is of specific interest for the NW-IGP as it is strongly affected by a seasonal source in the form of wheat residue burning. Crop residue burning activity from the NW-IGP appear prominently in various fire count products such as MODIS or VIIRS fire counts. Our study provides the first in-situ observations which allow to assess whether VOC emissions from this pyrogenic source are properly represented in the available emission inventories.

**Changes in the manuscript:** We have inserted the following sentence at the end of the paragraph:

"The month of May is of special interest, as it is affected by widespread wheat residue burning in the NW-IGP. In the present study, we quantify the contribution of this important area source to the VOC burden at a downwind site. Our analysis includes several rarely reported nitrogen containing compounds which appear to have strong pyrogenic sources in this particular study region. Compounds such as amines, amides and isocyanic acid are presently not included in global emission inventories and the default atmospheric chemistry mechanisms, despite their potential importance for secondary aerosol formation and human health."

**Reviewer comment:** The aims of the paper need to be better outlined (e.g. in the last paragraph of the introduction the comparison with emissions inventories is not mentioned in the objectives).

**Changes in the manuscript:** We added the following at the end of the paragraph:

"We compare our source-receptor modelling output with several emission inventories such as REAS v2.1, EDGAR v4.3.2 and GAINS v5 to assess which emission inventory is most consistent with the results of our source receptor modelling study that employs in-situ observations."

3. METHODS - The description of methods should be revised in its content. In particular:

**Reviewer comment:** Why have authors chosen to use the US EPA PMF 5.0 model? A brief motivation and description of the model need to be provided along with relevant references.

**Changes in the manuscript:** We added the following brief description and motivation:

"The EPA PMF 5.0 receptor model (Paatero et al. 2014, Norris et al. 2014) is multivariate factor analysis tool (Paatero & Tapper 1994, Paatero 1997), which decomposes the data matrix $x_{ij}$ with i number of samples and j number of measured VOCs into two matrices, the factor contribution matrix $g_{ik}$ (which provides the mass g contributed by each factor to the individual sample) and the factor profiles matrix $f_{kj}$ (which provides the source profile/fingerprint of each individual source). Both matrices are established for a user defined number of sources p from the existing intrinsic variability in the dataset leaving behind a matrix of residuals $e_{ij}$.

$$x_{ij} = \sum_{k=1}^{p} g_{ik} f_{kj} + e_{ij}$$

The two primary advantages of the PMF over other source receptor modelling tools are its inherent non-negative constraints (Hopke 2016) and its capability of optimally weighing individual data points and assigning uncertainties which makes it possible to include less robust species that can be useful for defining real sources. The EPAv5.0 model is superior when compared to other source receptor modelling tools as contains advanced rotational features (Paatero & Hopke) which allow to constrain the rotational ambiguity in a manner that pushes the PMF solution toward the real world space."

**Reviewer comment:** Almost the entire part of the methodology in Section 2.4 and 2.5 is left to the supplement or to other studies. Since it is a fundamental part of the methodology used in this the study, I would suggest to expand these sections. On the other hand, the detailed description in section 2.2. is not really relevant for this study, and should be cut/shortened.

**Author response:** We have expanded section 2.4 and 2.5 in the main manuscript and removed the relevant sections from the supplement. We have also shortened section 2.2 but retained the technical details of how the input data was generated.

**Changes in the manuscript:** Page 3 line 6 was shortened to:

"
As described in greater detail in Sinha et al.,( 2014),
ambient […]"
Section 2.4 was expanded to:
"**2.4 Conditional Probability Function analysis**
We perform a conditional probability function (CPF) analysis (Leuchner and Rappenglück,
2010) which aids in identifying physical locations of different PMF source factors without
using back trajectories (Xie and Berkowitz, 2006). The CPF is computed using the factor
contribution of the PMF model in combination with the wind direction at the receptor site. It
quantifies the probability of factor contributions surpassing a certain threshold (75[th]
percentile) for a particular wind direction sector thereby highlighting directional dependency
of source factors and is defined as follows:

$$CPF = \frac{m_{\Delta\theta}}{n_{\Delta\theta}} \qquad (2)$$

[revised manuscript text omitted]

**Reviewer comment:** 4. RESULTS AND DISCUSSIONS -
Overall results are too descriptive, and there are repetitions of information that figures already provide. I suggest to focus more on what can be deduced from the analysis rather than on its description.

**Author response:** We appreciate this advice by the anonymous reviewer #1 and have restructured our results and discussion section. The former section 3.1 has been combined with some details regarding the model output validation which were spread out over sections 3.2-3.6 and has been shifted to a new section "2.4 Validation of the PMF output" in response to one of the comments of the anonymous reviewer #2. Our Results and discussion section now starts with the content of the former section 3.7 (now shifted to 3.1) "Split up of VOC Emission Sources in Mohali and their contribution to Ozone and SOA Formation Potential". Sections 3.2-3.6 containing the description of the PMF results for the individual factors have been re-written to focus on what the analysis means rather than on describing the results.

**Changes in the manuscript:**

**Section 3.2 now reads:**

[revised manuscript text omitted]

**Reviewer comment:** • Section 3.8 presents the comparison between the source apportionment study and emission inventories estimates, i.e. a point vs gridded data. Is it sufficient to filter gridded data for LAT LONG from which air mass trajectories reach the site within one day to make the comparison reliable?

**Author response:** Air is a rapidly moving medium, in particular in May when the average wind speed is 5.6 ms$^{-1}$. Hence, the comparison of a receptor point with a much larger gridded area of an emission inventory should not be a concern. In fact, Sofowote et al. 2015 (Atmos. Environ. 108:151–57) used the PMF to source apportion the impact of distant sources on the

PM$_{2.5}$ aerosol burden at 5 remote locations in Ontario, Canada. We think that the more pertinent question is: How large should that gridded area be for a meaningful comparison? Many of the very specific tracers have short photochemical lifetimes of less than a day (e.g. styrene, C-8 and C-9 aromatics). Since these short lived compounds feature prominently in several source profiles, rather than being absent, this indicates that e.g. the 4-wheeler emissions on average have been subjected to photochemical aging for less than 4-10 hours prior to reaching the site. On the other hand, other compounds e.g. toluene (2 days), benzene (6 days) or acetonitrile (months) could have been transported much further away. The wheat residue burning source shows the greatest cross correlation for a lag time of 2 days indicating that emissions from distant sources can and do impact the site with a time lag. Hence we chose a compromise between the two sets of compounds in terms of lifetimes and delineated a fetch region of 1 day for the comparison with the emission inventories. This fetch region includes the areas where the maximum number of wheat residue burning fire counts are observed by satellites while avoiding a size that is too large to be consistent with the relatively unaltered signature of some of the other PMF source profiles.

**Changes in the manuscript:** We have inserted the following text into the newly created section 2.6

"This filtering is required because compounds with photochemical lifetimes of less than a day (e.g. styrene, C-8 and C-9 aromatics) feature prominently in several source profiles indicating that most of the transport sector emission were less than a day old when they reached the receptor site. Other compounds with longer lifetimes such as toluene (2 days), benzene (6 days) or acetonitrile (months) can reach the site from more distant sources. The wheat residue burning source shows the greatest cross correlation for a lag time of 2 days indicating that emissions from distant sources can and do impact the site with a time lag. The fetch region chosen for comparison with the emission inventories includes the areas where the maximum number of wheat residue burning fire counts are observed by satellites while avoiding a size that is too large to be consistent with the relatively unaltered signature of some of the other PMF source profiles. "

**Reviewer comment:** Moreover, the study considers May 2012, while emissions inventory data are for 2008/2010. Which are the uncertainties in using these approaches in the comparison? Authors should justify and better describe these choices.

**Author response:** We have reduced the uncertainties of the comparison by switching from EDGARv4.2 to the more recent version 4.3.2 for the year 2012. As far as REASv2.1 for the year 2008 is concerned, we could not improve the comparison as the NMVOC dataset of the MIX Asia 2010 inventory is identical to the NMVOC dataset of the REAS 2008 inventory. When it comes to the uncertainties introduced by comparing one month's data with an annual average emission inventory is concerned there are two parts to the answer.

1)The first part of the answer is that at present the only inventory that gives monthly data is in no way better than the inventories which provide only annual average data as the monthly data hardly differs from the sum of annual emissions divided by 12. Methane emissions from rice paddies in Punjab persist in the REAS emission inventory throughout the year even in months in which rice is not grown. Other sources do not appear to have been treated differently. Hence de facto there is no seasonality in any of the emission inventories available at present, a short coming that must be overcome in the long run but is beyond the scope of this work.

2) For emission inventories that do not provide monthly data, we have facilitated the comparison of the PMF output of the month of May which is affected by a strongly seasonal source (crop residue burning). To do so, we calculate hypothetical pie charts which attribute annual crop residue burning emissions over the region only to the 2.5 months when crop residue burning actually occurs (middle of October to end of November and May). This should reduce the uncertainty of the comparison. It allows to assess whether the model has the correct annual total emissions of the crop residue burning source and just lacks the proper distribution in the form of monthly data or is off with respect to the total annual emissions itself.

**Changes in the manuscript:** The following two text segments have been included in section 2.6

"Annual emissions were available for EDGAR (2012) and GAINS (2010), whereas, REAS provided monthly data (May 2008). However, Figure S6 shows that despite providing monthly data, the REAS emission inventory has very little seasonality for any of the sources."

"To facilitate the comparison of the PMF output of the month of May which is affected by a strongly seasonal source (crop residue burning) with emission inventories that provide only annual data, we calculate hypothetical pie charts which attribute annual crop residue burning emissions over the region only to the 2.5 months when crop residue burning actually occurs (middle of October to end of November and May)."

Figure 8 has been changed – so has the accompanying text.

[Figure]

Figure 8 has been revised and now includes EDGAR v4.3.2 (2012) instead of v4.2 (2008) and have updated the discussion accordingly. The latest EDGAR represents a significant improvement over the EDGAR HTAP and v4.2.

We have also added supplementary figures to compare speciated emission inventories with the PMF output for individual aromatic compounds

[Figure]

**Figure S8a:** Comparison of the PMF output with benzene emission inventories for the study region.

[Figure]

**Figure S8b:** Comparison of the PMF output with toluene emission inventories for the study region.

[Figure]

**Figure S8c:** Comparison of the PMF output with xylenes in the emission inventories for the study region.

[Figure]

**Figure S8d:** Comparison of the PMF output of C-9 aromatic compounds with the class "other aromatic compounds" in the emission inventories for the study region.

**Reviewer comment:** 5. CONCLUSIONS - It would be more valuable for the reader if the authors focused more on the achievements and implications of the results. The last paragraph of 3.8 may be included in the conclusions rather than in results.

Done we have shifted the paragraph and have re-written the conclusions. It now reads as follows:

"Our results highlight that for accurate air quality forecasting and modelling it is essential that emissions are attributed only to the months in which the activity actually occurs. This is important both for emissions from crop residue burning (which occur in May and from Mid-October to the end of November). Annually averaged emissions are unlikely to yield accurate air quality forecast in regions affected by such seasonal events. At present, more specialized fire emission inventories such as FINN (Wiedinmyer et al., 2011) must be used to account for the full seasonality and day to day variations of open burning emissions. We also demonstrate, that the source profiles obtained as PMF output can be validated and matched against samples collected at the potential sources to validate the factor identification.

For the human class I carcinogen benzene, the traffic factor alone contributed to 47% of the total benzene mass at this receptor site followed by residential biofuel use and waste disposal (25%) and industrial emissions and solvent use (20%). This stands in stark contrast to various emission inventories which consider domestic biofuel usage (39%), agricultural residue burning (19%) and industries (24%) to be the most important sources of benzene emissions. Since the annual NAAQS for benzene is exceeded at this receptor site (Chandra and Sinha, 2016), all three sectors must be targeted for emission reductions.

For the emerging contaminant isocyanic acid, photochemical formation from precursors (37%), wheat residue burning (25%) and biofuel usage and waste disposal (18%) were the largest contributors to human exposure. The monthly average isocyanic mixing ratio of 1.4 ppb exceeds concentrations that can, after dissociation at blood pH, result in blood cyanate ion concentrations (Roberts et al., 2011) high enough to produce significant health effects in humans (Wang et al., 2007) such as atherosclerosis, cataracts and rheumatoid arthritis due to protein damage. Peak mixing ratios of this compound exceed 3 ppb in some night time wheat residue burning plumes. Wheat residue burning was also the single largest source of the photochemical precursors of isocyanic acid, namely, formamide, acetamide and propanamide, indicating that this source must be most urgently targeted to reduce human concentration exposure to isocyanic acid.

Overall it appears that none of the emission inventories is ideal at the present. Our PMF solution suggests that transport sector emissions may be underestimated by GAINSv5.0 and EDGARv4.3.2, while the combined effect of residential biofuel use and waste disposal emissions as well as the VOC burden associated with solvent use may be overestimated by all emission inventories. Agricultural waste burning emissions of some of the detected compound groups (ketones, aldehydes and acids) are missing in the EDGARv4.3.2 inventory while aromatic emissions from the same source appear to be overestimated. Thus, large improvements are required in existing emission inventories for correct source attribution and inclusion of missing compounds over this densely populated region of the world."

**3 Minor comments**

**Reviewer comment:** 1. First author name (Pallavi) is missing.

Pallavi is a single name author. Her orcid is https://orcid.org/0000-0003-3664-6260

**Reviewer comment:** 2. Page 2 line 5 '...deserve further study' this sentence need citation.
**Author response:** This sentence refers to the previous sentence. Citations have been added to the previous sentence (Pawar et al. 2015, Sinha et al. 2014, Kumar et al. 2016)

**Reviewer comment:** 3. Page 2 line 31 '...and strong photochemistry' this sentence need citation.
**Author response & changes in the manuscript:** A citation to Sinha et al. 2014 has been added

**Reviewer comment:** 4. Section 2.3: need to add cross references to Table S3, Figure S4 a, b c.

**Author response:** done, we have added the cross reference in line 26 page 3

Figures S4 a, b c show how the factor profile, percentage of each VOC originating from a certain source, and the factor contribution change while increasing the number of factors in the model.

and line 1 page 4

A list of the constraints applied is provided in the supplementary table S3

**Reviewer comment:** 5. Page 9 line 16 'However, Figure S5..'. It is Figure S6 in the Supplement.

**Author response:** we have changed the numbering of several figures in the supplement as Reviewer #2 asked us to include an additional plot. The numbers are now consistent with the numbering in the manuscript.

**Reviewer comment:** 6. Figure 1 (b): add lat - long grid. It may be worth to add in the caption the exact coordinates of the site.

**Author response:** We have added the exact coordinates of the site instead.

We don't agree that adding a grid to the bottom figure is a good idea. It becomes a mess since Google Earth does not seem to allow us to define the grid spacing. It doesn't even seem to allow us to choose a different font size for the location labels and the grid labels. We are dealing with an area of less than 1 x 1 degree, so the figure with grid on looks ugly.

Figure 1b with grid on:

[Figure]

**Other minor corrections:** While preparing the new supplementary Figure S7 a small mistake in the calculation of the factor time series in $\mu g/m^3$ was spotted and corrected in Figure 5,7, S5c and throughout the manuscript.

**Response to Anonymous Referee #2**

**Reviewer comment:** The article titled "Source apportionment of volatile organic compounds in the northwest Indo–Gangetic Plain using positive matrix factorisation model" by Pallavi et al., is generally well written and contains some useful information. VOC source apportionment studies are sparse in India, and this study could encourage more such studies in future which is required to understand the VOCs impact on air quality. However, on many occasions, authors seem to over-interpret the results and have drawn some rather farfetched conclusions. I would recommend publication provided my concerns are being addressed satisfactorily.

**Author response:**
We thank the anonymous reviewer for his/her critical feedback and have addressed the comments individually as detailed below.

Abstract:

**Reviewer comment:** Numbers can be presented in a better way for ease of reading. Authors can put the percentage contribution of different factors/parameters in the parenthesis beside them.

**Author response:** We thank the anonymous reviewer for this valuable suggestion. The anonymous reviewer #1, suggested to focus the abstract more on the big picture. In response to both comments we have reduced the numbers in the abstract and added the percentage contribution of different factor/parameters in parenthesis beside them. It now reads as follows:

**Changes in the manuscript:**
"In this study we undertook quantitative source apportionment for 32 volatile organic compounds (VOCs) measured at a suburban site in the densely populated North-West Indo-Gangetic Plain using the US EPA PMF 5.0 Model. Six sources were resolved by the PMF model. In descending order of their contribution to the total VOC burden these are "biofuel use and waste disposal" (23.2%), "wheat-residue burning" (22.4%), "cars" (16.2%), "mixed daytime sources" (15.7%), "industrial emissions and solvent use" (11.8%) and "two-wheelers" (8.6%).

Wheat residue burning is the largest contributor to the total ozone formation potential (32.4%). For the emerging contaminant isocyanic acid, photochemical formation from precursors (37%) and wheat residue burning (25%) were the largest contributors to human exposure. Wheat residue burning was also the single largest source of the photochemical precursors of isocyanic acid, namely, formamide, acetamide and propanamide, indicating that this source must be most urgently targeted to reduce human concentration exposure to isocyanic acid in the month of May. Our results highlight that for accurate air quality forecasting and modelling it is essential that emissions are attributed only to the months in which the activity actually occurs. This is important for emissions from crop residue burning (which occur in May and from Mid-October to the end of November).

The SOA formation potential is dominated by "cars" (36.9%) and "two-wheelers" (22.1%), which also jointly account for 47% of the human class I carcinogen benzene in the PMF model. This stands in stark contrast to various emission inventories which estimate the transport sector contribution to the benzene exposure as (~10%) and consider residential biofuel use, agricultural residue burning and industries to be more important benzene sources. Overall it appears that none of the emission inventories represent the regional emissions in an ideal manner. Our PMF solution suggests that transport sector emissions may be underestimated by GAINSv5.0 and EDGARv4.3.2 and overestimated by REASv2.1, while

the combined effect of residential biofuel use and waste disposal emissions as well as the VOC burden associated with solvent use and industrial sources may be overestimated by all emission inventories. Agricultural waste burning emissions of some of the detected compound groups (ketones, aldehydes and acids) are missing in the EDGARv4.3.2 inventory."

**Reviewer comment:** Methods:
Sec 2.3: Line 20, why 20%? Please explain and incorporate in the manuscript as well.
**Author response:** We chose to assign 20% uncertainty to all masses to avoid a situation where the difference in the assigned uncertainty drives the PMF to dedicate a disproportionate number of a factors towards minimizing Q of a few compounds at the expense of others which may be equally useful as tracers of specific activities. The lower reported uncertainty of some compounds (8-12%) in Sinha et al. 2014 can be primarily attributed to the fact that the instrument has been calibrated with more than one independently sourced calibration gas bottles for that particular compound and the fact that the respective m/z has a good signal to noise ratio and high signals. For some other compounds the sensitivity had to be derived from theory, because no calibration gas is available, hence they carry a larger error.

We have followed the advice of Paatero et al. 2014, Atmos. Meas. Tech., 7, 781–797 and performed sensitivity studies to better understand how errors and their handling can impact the PMF output in our specific case.

In our specific case the fact that toluene has one of the smallest reported measurement errors (8.6 % in Sinha et al. 2014) in combination with the fact that there is a genuine and abundant source with a normalized source profile that is dominated by toluene (tailpipe exhaust of petrol fuelled 2-wheelers) can result into a serious modelling error. This problematic behaviour is observed for this particular dataset, because the second most abundant compound in the same tailpipe exhaust source profile (the xylenes) carries a larger uncertainty (11.8 %) and can be accommodated in other source profiles with a smaller penalty on Q. Most real world traffic contains a mixture of 4-wheelers and 2-wheelers and the ratio of these two vehicle classes in the traffic varies as a function of air mass origin and time of the day. At the same time the benzene/toluene ratio of all aged plumes varies with the photochemical age of the air mass. When all these factors are combined the situation becomes such that while running the model with differential errors the lowest Q for the equation

$$Q = \sum_{i=1}^{n} \sum_{j=1}^{m} \left[ \frac{x_{ij} - \sum_{k=1}^{p} g_{ik} f_{kj}}{u_{ij}} \right]^2$$

is obtained by creating a separate toluene factor and removing toluene from the factor profile of all combustion source profiles. In other words, in this specific dataset assigning correct but different random errors to different m/z triggers a serious modelling error which appears already in a 4 factor solution and is retained through any higher number of factors. Assigning equal random errors to all m/z prevents this modelling error from occurring. Hence, we assigned an error of 20% to all masses even though in reality only few strong m/z ratios (formaldehyde, propyne, styrene and phenol) carry such a large error. The reviewer is, however, correct that this choice of using the largest error for all compounds is somewhat arbitrary and one could instead use the average uncertainty of all the strong compounds errors

(i.e. assigning ~10% uncertainty to all compounds would also prevent the modelling error). The magnitude of the chosen error will impact the magnitude of Q (which will increase by a factor of ~4 when 10% instead of 20% uncertainty is assigned) but will not change the model output as long as equal uncertainty is assigned to all masses. However, considering the accuracy and precision error while initializing the PMF may not actually be the right approach at all, considering that the software treats the errors as random. One could argue that only the precision error should be considered while assigning errors in the software. However, as long as equal uncertainty is assigned to all strong m/z the assigned uncertaintyt will not change the model output and conclusions.

**Changes in the manuscript:** We have incorporated this reason and the section now reads "All 32 species were assigned a fixed 20% in the uncertainty, which represents the largest uncertainty estimated for strong compounds, to avoid a situation where the difference in the assigned uncertainty drives the PMF to dedicate a separate factor towards minimizing Q of a single compound with low uncertainty (toluene) by taking it out of all other source profiles and opening a separate factor profile containing just a single compound."

**Reviewer comment:** Line 20, more than 50% of the measured species (18 of 32) are weak, isn't that going to influence the robustness & reliability of the PMF output?
**Author response:** This is definitely going to impact the robustness and reliability of the PMF output in a positive manner. A poor signal to noise ratio indicates that the measured values of a species throughout most of the time series are very close to the detection limit. All instruments have a higher precision error close to the detection limit. This is why the manual recommends assigning masses with low S/N ratio "weak" and we have followed this instruction for all compounds that have a poor S/N ratio and do not show any strong peaks. However, in our opinion S/N ratio should not be blindly used as a criterion to make masses weak. Let us consider the hypothetical scenario of a compound emitted only by a single source impacting the site. There can be a situation where such a source impacts the site only rarely (say less than 5% of the time) but when it does the plume brings a very high concentration of that tracer compound. In such a case that specific tracer could be extremely precious for constraining the rotational ambiguity of the PMF solution, even though its average S/N ratio would be very poor (because 95% of the values in the time series are noise around the detection limit). Hence one always needs to look at every species of the input dataset carefully to assess whether it should be made weak just because of its S/N ratio.

There can be reverse cases of masses with a high S/N ratio (for which the average concentration is always far above the detection limit throughout the time serious), which can negatively impact the PMF rotational ambiguity when not labelled as weak. This is the case for all masses with potential isobaric interferences. Let us consider an m/z where one of the compounds falling onto the mass to charge ratio is of pyrogenic origin and the other one a tracer for biogenic emissions or a product of daytime photochemistry and discuss how this will impact the PMF model output depending on whether the species is a strong or weak species. Any peak in the concentration observed can be due to either of the contributors i.e. due to a combustion source alone or due to biogenic emissions/daytime photochemistry alone or due to a mixture of both. The most serious impact of this on the model performance is that it can make resolving the rotational ambiguity difficult and can cause modelling errors. Resolving rotational ambiguity requires that the matrix contains a sufficient number of zero values where a source is totally absent. When two sources with different temporal profiles (night-time combustion and daytime biogenic emission or night-time combustion and daytime photochemistry) contribute different compounds to the same m/z ratio, zero values

are almost absent in that particular column of the matrix. When this column is made "weak" and given a higher uncertainty, other "strong" tracers with genuine zero values and strong peaks that can be attributed to a specific sources define source profiles and this reduces the rotational ambiguity of the model. The "weak" compounds with isobaric interferences are distributed among the source profiles available as per the solution that minimizes Q but they do not define any of the profiles. In our opinion, this is the most appropriate way to treat m/z ratios with potential isobaric interferences. As already described in the supplement and the main text we have made such masses weak in the PMF to improve the quality of the PMF output.

**Changes in the manuscript:** We have added a clarification that this makes the model more robust.

"Designating sources with isobaric interferences as weak is warranted because when two sources with different temporal profiles (night-time combustion and daytime biogenic emission or night-time combustion and daytime photochemistry) could potentially contribute different compounds to the same m/z ratio, zero values are almost absent in that particular column of the matrix and the tracer is affected by additional uncertainty not appropriately expressed by merely looking at the instrumental measurement error and the signal to noise ratio. When this column is made "weak" and given a higher uncertainty, other "strong" tracers, representing a single compound, define source profiles and this reduces the rotational ambiguity of the model. The "weak" compounds with isobaric interferences tend to be distributed among the source profiles available as per the solution that minimizes Q but they do not define any of the profiles."

**Reviewer comment:** Line 24, Why the authors chose to remove missing values instead of replacing them with some other values as mentioned in the literature? Is this the standard practice? what is the % of missing values in the total sampled points?

**Author response:** Replacing missing values with the median while assigning it a greater uncertainty in the PMF helps a lot when the PMF is run with different tracers measured with different sets of instruments, each of which has a different set of missing values. The default setting of the EPA PMF model described in the literature was developed for such a scenario. To illustrate let us consider using a dataset with data from 10 different instruments each of which individually has less than 10% downtime in a situation where unfortunately problems rotate. In that scenario for > 50% of the data points a few variables would be missing. Using the exclude missing value option in such a case would mean throwing out more than half of the dataset as the model removes all lines (=points in time) with a missing value, even if only a single column has a missing value. In such cases lines with missing values still have a lot of data (because one instrument is down the other instruments are running) and only a small subset of species is missing for each point to be potentially excluded. Hence the default model setup suggests filling in missing values with the median of the time series while assigning a greater uncertainty to that point.

However, we are dealing with measurements of a single instrument and <5% of missing values in a month. Filling missing values does not improve the quality of the model output in our case. When the PTR-MS is undergoing calibration or ion source cleaning, there is no ambient data at all available for the gap. Hence the gap filling is unnecessary. It serves no purpose and would hardly affect the model output as all parameters would be filled in with their respective median for that particular point in time.

**Changes in the manuscript:** added ( <5%) after "missing values"

**Reviewer comment:** Sec 3.3

Line 1-3, R= 0.4 is not a good correlation, at best it can be termed as moderate. Please rewrite the explanation on why fire count is the best tracer for factor 2.

**Author response:** With best we simply meant that the R was better than that of other potential independent tracers such as NOx (which correlated more with transport sector emissions) and CO (which correlated best with the more regular open burning activities such as biofuel use and waste disposal). However, we understand now, that this could be misunderstood and have revised the sentence.

**Changes in the manuscript:** "Figure 3 shows that the factor profile correlates most strongly with flaming wheat residue burning (R=0.9). The average contribution of wheat residue burning to the total NMVOC mass at the receptor site and the daily fire counts over the NW-IGP show a moderate cross correlation of R=0.4 with a lag of 2 days (Figure 5)."

**Reviewer comment:** Sec 3.7
Line 7, in PM2.5, 2.5 should be subscript
**Author response:** Done this section has become Sec 3.1 in the restructured manuscript

**Reviewer comment:** Line 7, I don't think the way SOA being calculated enable the authors to make such strong quantitative assertion about the SOA contribution to PM2.5 in Mohali. At best, the adopted method can provide a qualitative and comparative assessment of SOA production efficiency among different PMF factors. I would suggest to remove or modify. line 6-8 to reflect this.

**Author response:** We understand that the SOA formation potential as calculated has its limitations and depends on the $NO_X$ regime and may even show a non-linear dependence on VOC and NOx concentration for some compounds (Xu et al., 2015, Atmospheric Environment 101, 217-225). However, we believe that providing a boundary condition may be useful. We have modified lines 6-8. We now explicitly mention that we applied the "SOA yields for the low NOx regime" in the relevant section of the materials and methods, which the reviewer #1 asked to extend and in this section. We also now put the calculated SOA formation potential (i.e. the ~17 μg/m3) in brackets behind its first mention in the paragraph as we believe that despite all short comings this number provides an important perspective. In support, we have also qualified the estimate by citing more studies.

**Changes in the manuscript:**
"While the calculated SOA formation potential particularly from transport sector emissions (Ensberg et al., 2014) and aromatic compounds (Li et al., 2017, Li et al., 2018) is affected by large uncertainties and may depend in a non-linear fashion on NOx and VOC concentrations (Xu et al 2015) our calculated SOA formation potential seem to indicate that SOA formation could contribute significantly to the overall PM2.5 burden (104 μgm-3).

**Reviewer comment:** Sec 3.8
I am not sure about the utility or purpose of this section. Are authors trying to use this comparison as another tool for PMF results validation? Or to suggest which inventory is better? Every emission inventory is developed based on some underlying assumptions and approximations. I would rather be very surprised if a single site based study can reproduce or match the emission inventory values. It is quite expected that differences will be there and even a perfect match doesn't necessarily validate emission inventories or the PMF results, especially in a complex source environment as in India. Several assumptive statements were made to explain the mismatch/less match between PMF results and emission inventory values. So, based on this comparison one can't really assert which inventory is better or more representative than others. Authors should remove or rephrase the section to reflect those concerns.

**Author response:** This section is meant to identify which of the currently used emission inventory represents the regional sources best. This is the major motivation behind any source-receptor modelling study. The anonymous reviewer is correct that every emission inventory is developed based on some underlying assumptions and approximations. Some of these assumptions and approximations can be awfully wrong and the purpose of source receptor modelling studies is to point out such discrepancies. For example, several PMF based source receptor modelling studies in Europe found that the solvent source could be overestimated in most emission inventories while the transport sector may be underestimated (Gaimoz et al. 2011 *Environ. Chem.* **2011**, *8*, 91–103., Niedojadlo et al. 2007 *Atmos. Environ.* **2007**, *41*, 7108., Lanz, et al. 2008 *Atmos. Chem. Phys.* **2008**, *8*, 2313.). Such discrepancies between inventories and source receptor modelling results which got replicated in several studies in different countries ultimately triggered a new series of road tunnel studies and on-road emission factor measurements to re-evaluate the assumptions and approximations used while building the transport sector emission inventories. These efforts not only resulted in a significant upward revision of transport sector emission estimates for NMVOCs while shifting from the EDGARv4.2 inventory for the year 2010 to EDGARv4.3.2 for the year 2012 but also exposed the diesel cheat software that switched off pollution control devices when the vehicles were driving on the road. Therefore, we believe that reality checks bases on source receptor modelling of ambient data perform an important role. Their potential significance is even larger in a complex environment where activity data for informal sector industries and activities that officially don't happen (e.g. open waste burning) is hard to obtain while at the same time proper emission factor measurements for many sources are lacking. We, therefore, insist that this section is important to retain.

The validation of PMF results in all prior studies has been performed by cross correlating one or several columns of one of the two matrices produced during the factor decomposition, namely the factor contribution matrix, with independent variable in the form of the time series of compounds that were not used to drive the model. We performed this cross verification step for all six identified factors using the species NOy (cars & 2-wheelers), $SO_2$ (industrial emissions), CO (domestic fuel usage and waste disposal), fire counts (wheat residue burning) and $O_3$ (mixed daytime factor). However, our study, to the best of our knowledge, is the first one to add an additional verification step in the form of grab samples collected at the source which were used to independently verify the factor profiles (i.e. the second matrix) that the PMF model created during the matrix decomposition. This validation was performed using samples collected at the source for five of the six factor profiles (wheat residue burning, domestic fuel usage and waste disposal, industrial emissions and solvent use, car tailpipe emissions, and 2-wheeler tailpipe emissions). It appears that this validation procedure was not described clearly enough, hence we have now added a section describing the procedure to the materials and methods section. We added a section 2.4 Validation of the PMF output. Some of the text in this section has been shifted from section 3.1 to this section. We also added a new reference since the source signature of brick kilns has recently been published and has now been included.

We have also switched to a new version of EDGAR (v4.3.2) which has recently become available and have added more depth to the comparison by looking at individual compound classes of the speciated emission inventory rather than just at the total VOC mass. We have also removed some of the quantitative statements.

**Changes in the manuscript:**
**"2.4 Validation of the PMF output**

[revised manuscript text omitted]
. ~~is characterized by elevated concentration levels of benzene (1.4 µg m$^{-3}$), toluene (2.3 µg m$^{-3}$), sum of C-8 aromatics (3.5 µg m$^{-3}$) and sum of C-9 aromatics (2.7 µg m$^{-3}$) and explained 35%, 30%, 53% and 58% of the total benzene, toluene, C-8 aromatics and C-9 aromatics mass in the PMF model, respectively.Features of car's factor profile also resemble gasoline evaporation headspace for diesel (R=0.5) collected at a petrol pump. This indicates that the factor profile consiststoluene to benzene ratio of this profile (1.4) is typical for traffic emissions (1.5-2.3) (Som et al., 2007; Hoque et al., 2008; Chandra et al., 2018) and 
[revised manuscript text omitted]
 and domestic biofuel use and waste disposal. REAS overestimate the contribution of industrial activity and solvent use in the month of May (22%). Our PMF solution for road transport sector emissions (30.5 %) lies in between the estimates of GAINS (558 $Gg\,y^{-1}$, 24 %) and REAS (1230 $Gg\,y^{-1}$, 36.2 %), possibly, because not all pre-2000 super-emitters for which the 20-year vehicle lifetime has been exceeded have been retired as planned.

Overall it appears that GAINS, the emission inventory with the lowest absolute emissions from residential and commercial biofuel use shows the best agreement with our PMF solutionnone of the emission inventories is ideal at the present. Our PMF solution suggests that transport sector emissions may be are underestimated by approximately a factor of 1.5 in GAINS and EDGARv4.3.2, while the combined effect of residential biofuel use and waste disposal emissions as well as the VOC burden associated with solvent use may be overestimated by a factor of 1.3 in the same all emission inventoryies. Similar results have been reported previously. Sarkar and co-workers (Sarkar et al., 2017) reported an underestimation of transport sector emissions for the REAS and EDGAR emission inventory for the Kathmandu valley in Nepal and an overestimation of the residential biofuel use and waste disposal source in all emission inventories, while Gaimoz and co-workers (Gaimoz et al., 2011) reported an overestimation of the VOC emissions from solvent use in Paris.

REAS and EDGAR overestimated residential bio fuel usage emissions even more than GAINS. EDGAR underestimated transport sector emissions and industrial emissions and solvent usage while REAS overestimates the importance of the same two sources. REAS also fails to include agricultural residue burning as a source.

Our results highlight that for accurate air quality forecasting and modelling it is essential that emissions are attributed only to the months in which the activity actually occurs. This is important both for emissions from crop residue burning (which occur in May and from Mid-October to the end of November) and emissions from wildfires (which are restricted to the dry season and peak in April and May). Annually averaged emissions are unlikely to yield accurate air quality forecast in regions affected by such seasonal events. At present, more specialized fire emission inventories such as FINN (Wiedinmyer et al., 2011) must be used to account for the full seasonality and day to day variations of open burning emissions. We also demonstrate, that the source profiles obtained as PMF output can be validated and matched against samples collected at the potential sources to validate the factor identification.

We find that the GAINSv5.0 emission inventory for the year 2010 agreed best with the in-situ data derived PMF solution for May 2012.

**4 Conclusions**

Our results highlight that for accurate air quality forecasting and modelling it is essential that emissions are attributed only to the months in which the activity actually occurs. This is important for emissions from crop residue burning (which occur in May and from Mid-October to the end of November). Annually averaged emissions are unlikely to yield accurate air quality forecast in regions affected by such seasonal events. At present, more specialized fire emission inventories such as FINN (Wiedinmyer et al., 2011) must be used to account for the full seasonality and day to day variations of open burning emissions. We also demonstrate, that the source profiles obtained as PMF output can be validated and matched against samples collected at the potential sources to validate the factor identification. Six VOC emission sources were extracted via PMF simulations from the dataset comprising of 32 VOC species measured online at primary temporal resolution of 1 minute at a sub-urban site in Mohali in the summer of 2012. US EPA PMF 5.0 Model was used for source apportionment of VOCs and PMF-resolved factors included traffic exhaust, biofuel use and waste disposal, wheat-residue burning and mixed daytime sources (comprising of biogenic emissions and photochemical formation), industrial emissions and solvent use, which along with the residuals,accounted for 25.1%, 23.2%, 22.4%, 15.7%, 11.8% and 1.7%, respectively, of the total VOC mass concentration.

[revised manuscript text omitted]

**Figure S4** $Q/Q_{exp}$ plot with increasing number of factor. The absolute Q is relatively low indicating that it may be more appropriate to only consider the 10% precision error of the PTR-MS instead of including the accuracy error while specifying the uncertainty in the PMF input. However, since equal uncertainty was applied to all strong m/z this only affects the absolute Q value and not the model output.

[Figure]

**Figure S5a.** Evolution of PMF factor profiles from 3 to 7 factor number solutions.

[Figure]

**Figure S5b.** Evolution of percentage contribution of different VOC species from 3 to 7 PMF factor solutions.

[Figure]

**Figure S5c.** Evolution of PMF factor contributions from 3 to 7 factor solutions.

[Figure]

**Figure S6.** REAS database comparison to VOC source sectors on monthly and yearly resolution scales.

[Figure]

**Figure S7.** Time series of the total mass contributed by the different sources to the overall VOC mass

[Figure]

**Figure S8a:** Comparison of the PMF output with benzene emission inventories for the study region.

[Figure]

**Figure S8b:** Comparison of the PMF output with toluene emission inventories for the study region.

[Figure]

**Figure S8c:** Comparison of the PMF output with xylenes in the emission inventories for the study region.

[Figure]

**Figure S8d:** Comparison of the PMF output of C-9 aromatic compounds with the class "other aromatic compounds" in the emission inventories for the study region.

**References**

[revised manuscript text omitted]

---

## Author Response (AR2)

**Final Author response**

**Response to editor**

Comments to the Author:
Thanks for attending to the reviewer's comments. I think you have answered them adequately, but I do have a few comments on your response and the revised manuscript. I will note the page number but since you didn't provide line numbers, you may have to hunt to find the location.

Page 3, Kolkata is spelled two different ways.

Changed

Page 5. What is the parameter 'n' equation (3)?

Inserted "wherein n stands for the number of ozone molecules produces in the oxidation of $VOC_i$" before using n=2

Page 8. Does India use MTBE as a fuel additive?

Yes – India uses MTBE as an additive in petrol we now clarified this in the manuscript, according to the PetroIndustryNews dated 10th January 2019 MTBE capacity in India is expected to grow at a CAGR of 16.8% over the next four years.

Page 9, first paragraph. If two wheelers are more abundant in small towns, it seems like there should be some independent data on this.

Motor vehicles statistical yearbook India, which comes out annually has motor vehicle registration data. In 2015 (the most recent year for which data from all states is available) 2-wheelers made up 73.5% of the privately owned non-transport vehicles. State-wise raw registration data can be downloaded from http://mospi.nic.in/statistical-year-book-india/2018/189. The comparison of Union territories that are rich city-states such as Delhi (64% 2-wheelers) and Chandigarh (58% 2-wheelers) and UT's that are smaller city states (e.g. Pondicherry 86% 2-wheelers) or economically backward states like Uttar Pradesh (80% 2-wheelers) clearly reveals that car ownership is a distinctively urban and upper middle class phenomenon. In India poorest do not own personal means of transportation, those that are marginally better off own a bicycle (and know how to fit a family with 2-3 kids onto a normal gents' cycle) the lower middle class owns 2-wheelers (which can accommodate up to 3 adults and 3 kids if they must), car ownership is an upper middle class phenomenon.

We have inserted the following statement. "This is independently supported by vehicle registration data (http://mospi.nic.in/statistical-year-book-india/2018/189)."

Page 17. Response to reviewer's comments RE Section 2.3. It would be useful to note that relative uncertainties are most important here, as absolute uncertainties often cancel out among species measured by the same instrument.

We have inserted the following: "It should be noted that the precision error is more important for the PMF performance in this case, as the accuracy error often cancel out among species measured by the same instrument."

Page 18. End of paragraph 1, 'uncertainty'.

done

Page 18. Start of paragraph 4, 'serious' should be 'series'.

series

Figure 3. I can no longer distinguish the blue and green color bars in the top panel.

Color has been changed

References. There are a number of typos in this section, please review carefully.

done

**Response to Anonymous Referee #1**

**Reviewer comment:**
**1 Overall comment**
This study focuses on the source apportionment of VOCs measurements at a suburban site in the North-West Indo-Gangetic Plain. The period studied is the month of May 2012. Authors use a Positive Matrix Factorization Model (PMF) to resolve source contributions to VOCs, perform a conditional probability functional analysis to locate the different sources and calculate the ozone and secondary organic aerosols formation potential. Moreover, results of PMF are compared with the source apportionment of three different emission inventory estimates.
Overall, the analysis performed is interesting and valuable. However, the manuscript needs improvements in the logical framing of the work with respect to its contribution and implications to the field. Also the introduction and the results need to be improved in this sense. I recommend publication after the authors have addressed the following substantive concerns/comments on their manuscript.

**Author response:** We sincerely thank the reviewer for his encouragement and the in-depth comments and suggestions which have greatly improved the clarity of the manuscript and have helped us to emphasize the implications of this study to the field more clearly.
The detailed response to each comment and changes made in the manuscript are listed below.

**2 Major comments**

1. ABSTRACT - the abstract is a bit too technical. I recommend to focus more on the big picture and major findings and implication of the paper (as outlined in the conclusions).
**Author response:** We appreciate this feedback and have revised the abstract in accordance with it. Revised abstract reads as follows:

**Changes in the manuscript:**
"In this study we undertook quantitative source apportionment for 32 volatile organic compounds (VOCs) measured at a suburban site in the densely populated North-West Indo-Gangetic Plain using the US EPA PMF 5.0 Model. Six sources were resolved by the PMF model. In descending order of their contribution to the total VOC burden these are "biofuel use and waste disposal" (23.2%), "wheat-residue burning" (22.4%), "cars" (16.2%), "mixed daytime sources" (15.7%), "industrial emissions and solvent use" (11.8%) and "two-wheelers" (8.6%).
Wheat residue burning is the largest contributor to the total ozone formation potential (26.2%). For the emerging contaminant isocyanic acid, photochemical formation from precursors (37%) and wheat residue burning (25%) were the largest contributors to human exposure. Wheat residue burning was also the single largest source of the photochemical precursors of isocyanic acid, namely, formamide, acetamide and propanamide, indicating that this source must be most urgently targeted to reduce human concentration exposure to isocyanic acid in the month of May. Our results highlight that for accurate air quality forecasting and modelling it is essential that emissions are attributed only to the months in which the activity actually occurs. This is important both for emissions from crop residue burning (which occur in May and from Mid-October to the end of November).
The SOA formation potential is dominated by "cars" (36.9%) and "two-wheelers" (21.1%), which also jointly account for 47% of the human class I carcinogen benzene in the PMF model. This stands in stark contrast to various emission inventories which estimate the transport sector contribution to the benzene exposure as (~10%) and consider residential biofuel use, agricultural residue burning and industries to be more important benzene sources. Overall it appears that none of the emission inventories represent the regional emissions in an ideal manner. Our PMF solution suggests that transport sector emissions may be underestimated by GAINSv5.0 and EDGARv4.3.2 and overestimated by REASv2.1, while the combined effect of residential biofuel use and waste disposal emissions as well as the VOC burden associated with solvent use and industrial sources may be overestimated by all emission inventories. The agricultural waste burning emissions of some of the detected compound groups (ketones, aldehydes and acids) are missing in the EDGARv4.3.2 inventory."

**Reviewer comment:** 2. INTRODUCTION - the introduction should better frame the background of the study, its motivation and what is the new contribution of the work. In particular:
• Only one source receptor modelling study that has been cited is in the region of the study (Srivastava et al., 2005). Are there any source receptor modelling or more general studies that focus on VOCs over the IGP? If yes, they should be acknowledged. If no, this should be underlined.

**Author response:** We have cited all the source receptor modelling studies for VOC performed in India that are available in the peer reviewed literature. Beyond the studies we cited, there is one PMF study from the Kathmandu valley in Nepal (Sarkar et al. 2017) which represents a different environment and one attempted PMF study from the Eastern Himalayas (Sarkar et al.

2014, acpd-14-32133-2014), which did not make it into ACP. We have now made it more clear that no other VOC Source apportionment study in the IGP exists.

**Changes in the manuscript:** We inserted the following sentence after line 12 on page 2 "The only other source receptor modelling study in South Asia was conducted using a positive matrix factorisation model (EPA PMF5.0) with data collected in the Kathmandu valley, Nepal, as part of the SUSKAT campaign and attributed a negligible fraction of the anthropogenic VOC burden to residential biofuel usage (~14%). Instead different industrial sources including brick kilns (jointly 52%) and the transport sector (21%) were identified as the dominant VOC sources in the Kathmandu valley."

**Reviewer comment:** VOCs source apportionment estimates for the region under study are presented for different emissions inventories. However, it is claimed 'Considering the large discrepancies between bottom up inventories and different source receptor modelling studies', when 2/3 source receptor models studies presented so far are out of the understudied region. This claim need to be justified, or more appropriate studies need to be cited.

**Author response:** We agree with the reviewer that comparing performance of emissions inventories for the entire NW-IGP with a source receptor model study conducted in a specific megacity could introduce a bias. Since we have no other source receptor studies to fall back on, we have now made the comparison between the inventory and the source receptor modelling studies more specific and spatially accurate limiting it to the Delhi National Capital Region, Greater Mumbai and Greater Kolkata, when comparing with the source receptor modelling studies of these cities, respectively and have revised the text from line 5-11 accordingly.

In response to a later comment by the same reviewer about the mismatch between the emission inventory year and the year of our study we have now substituted EDGAR v4.2 with the latest EDGAR v4.3.2 version (Huang et al. Atmos. Chem. Phys., 17, 7683–7701, 2017)**,** with emissions for the year 2012, which appears to represent a significant improvement over the previous version. We have not included MIX Asia (Li et al. Atmos. Chem. Phys., 17, 935–963, 2017) since the NMVOC data of this inventory for India has been sourced from REASv2.1 without any changes. As a result of this update we have also revised the text of lines 11-17 in the introduction (not just the results and discussion section 3.8 and Figure 8).

**Changes in the manuscript:** The revised paragraphs now read:
"Previous source receptor modelling studies of VOC emission from India (Srivastava, 2004; Srivastava et al., 2005; Majumdar et al., 2009) produced results that conflicted strongly with the bottom up emission inventories, all of which contain significant emissions from residential fuel usage even when filtered for the New Delhi National Capital Region (41-45%), Greater Mumbai (32-36%) and Greater Kolkatta (33-59%). Transport sector emissions, according the bottom up emission inventories contribute only 15-35%, 17-43% and 6-14% to the total VOC emissions in New Delhi National Capital Region, Greater Mumbai and Greater Kolkatta, respectively. All previous studies from India employed a chemical mass balance (CMB) technique for ambient VOC source attribution and identified the transport sector as the main source of NMVOCs in the form of evaporative emissions (40-87%) in Mumbai (Srivastava, 2004), diesel internal combustion engines (26-58%) in Delhi (Srivastava et al., 2005) and roadway/refuelling exhaust (~40%) in Kolkata city (Majumdar et al., 2009). Except for the study performed in Kolkata which found a contribution of <10% from wood combustion, residential fuel usage was not identified as a potential VOC source in those source receptor modelling studies. The observed discrepancy could be partially caused by the fact a CMB is not necessarily an ideal tool for conducting source receptor modelling study in understudied environments as the model needs to be initialized with locally measured source profiles of all potentially significant sources. However, it is unlikely that this is the only

reason for the discrepancies between source receptor modelling outcomes and emission inventories.

Different bottom up emission inventories also have large discrepancies between each other when extracted for the NW-IGP. For our study region (27.4-34.9 °N and 72-79.8 °E), EDGAR v4.3.2 estimates that the road transport sector contributes only 18% of the total anthropogenic VOC emissions (440 Gg y$^{-1}$), while REAS v2.1 (and MIX Asia) attribute 35.8% of the total anthropogenic VOC emissions (1230 Gg y$^{-1}$) to this sector. For industrial emissions and solvent use, GAINS has the lowest (540 Gg y$^{-1}$) and EDGAR v4.3.2 the highest absolute emissions (900 Gg y$^{-1}$). Crop residue burning as VOC source is missing in REAS but accounts for a 6% (145 Gg y$^{-1}$) and 7% (163 Gg y$^{-1}$) share of the annual VOC emissions in EDGAR and GAINS, respectively."

**Reviewer comment:** The study takes into consideration a specific month, May 2012. It is needed to explain why this month is important for the region under study and which general conclusions can be made from it.

**Author response:** The month of May is of specific interest for the NW-IGP as it is strongly affected by a seasonal source in the form of wheat residue burning. Crop residue burning activity from the NW-IGP appear prominently in various fire count products such as MODIS or VIIRS fire counts. Our study provides the first in-situ observations which allow to assess whether VOC emissions from this pyrogenic source are properly represented in the available emission inventories.

**Changes in the manuscript:** We have inserted the following sentence at the end of the paragraph:

"The month of May is of special interest, as it is affected by widespread wheat residue burning in the NW-IGP. In the present study, we quantify the contribution of this important area source to the VOC burden at a downwind site. Our analysis includes several rarely reported nitrogen containing compounds which appear to have strong pyrogenic sources in this particular study region. Compounds such as amines, amides and isocyanic acid are presently not included in global emission inventories and the default atmospheric chemistry mechanisms, despite their potential importance for secondary aerosol formation and human health."

**Reviewer comment:** The aims of the paper need to be better outlined (e.g. in the last paragraph of the introduction the comparison with emissions inventories is not mentioned in the objectives).

**Changes in the manuscript:** We added the following at the end of the paragraph:

"We compare our source-receptor modelling output with several emission inventories such as REAS v2.1, EDGAR v4.3.2 and GAINS v5 to assess which emission inventory is most consistent with the results of our source receptor modelling study that employs in-situ observations."

3. METHODS - The description of methods should be revised in its content. In particular:

**Reviewer comment:** Why have authors chosen to use the US EPA PMF 5.0 model? A brief motivation and description of the model need to be provided along with relevant references.

**Changes in the manuscript:** We added the following brief description and motivation:

"The EPA PMF 5.0 receptor model (Paatero et al. 2014, Norris et al. 2014) is multivariate factor analysis tool (Paatero & Tapper 1994, Paatero 1997), which decomposes the data matrix $x_{ij}$ with i number of samples and j number of measured VOCs into two matrices, the factor contribution matrix $g_{ik}$ (which provides the mass g contributed by each factor to the individual sample) and the factor profiles matrix $f_{kj}$ (which provides the source profile/fingerprint of each

individual source). Both matrices are established for a user defined number of sources p from the existing intrinsic variability in the dataset leaving behind a matrix of residuals $e_{ij}$.

$$x_{ij} = \sum_{k=1}^{p} g_{ik} f_{kj} + e_{ij}$$

The two primary advantages of the PMF over other source receptor modelling tools are its inherent non-negative constraints (Hopke 2016) and its capability of optimally weighing individual data points and assigning uncertainties which makes it possible to include less robust species that can be useful for defining real sources. The EPAv5.0 model is superior when compared to other source receptor modelling tools as contains advanced rotational features (Paatero & Hopke) which allow to constrain the rotational ambiguity in a manner that pushes the PMF solution toward the real world space."

**Reviewer comment:** Almost the entire part of the methodology in Section 2.4 and 2.5 is left to the supplement or to other studies. Since it is a fundamental part of the methodology used in this the study, I would suggest to expand these sections. On the other hand, the detailed description in section 2.2. is not really relevant for this study, and should be cut/shortened.

**Author response:** We have expanded section 2.4 and 2.5 in the main manuscript and removed the relevant sections from the supplement. We have also shortened section 2.2 but retained the technical details of how the input data was generated.

**Changes in the manuscript:** Page 3 line 6 was shortened to:
 As described in greater detail in Sinha et al.,( 2014),  ambient [...]"

Section 2.4 was expanded to:
"**2.4 Conditional Probability Function analysis**
We perform a conditional probability function (CPF) analysis (Leuchner and Rappenglück, 2010) which aids in identifying physical locations of different PMF source factors without using back trajectories (Xie and Berkowitz, 2006). The CPF is computed using the factor contribution of the PMF model in combination with the wind direction at the receptor site. It quantifies the probability of factor contributions surpassing a certain threshold (75[th] percentile) for a particular wind direction sector thereby highlighting directional dependency of source factors and is defined as follows:

$$CPF = \frac{m_{\Delta\theta}}{n_{\Delta\theta}} \qquad (2)$$

Here, $m_{\Delta\theta}$ refers to number of samples exceeding the criterion value in a certain wind sector and $n_{\Delta\theta}$ counts the total number of data points in that respective wind sector. $\Delta\theta$ was assigned a value of 30°. "

Section 2.5 was expanded to:
"**2.5 Calculation of the ozone formation potential and SOA formation potential**
Ozone production potential for each of the PMF derived source factors was calculated based on the method used in Sinha et al., (2012) using the following equation:

$$Ozone\ production\ potential = \left(\sum_i k_{(VOC_i + OH)} [VOC_i]\right) \times [OH] \times n \qquad (3)$$

Here, n = 2 and [OH] = $10^6$ molecules/cm$^3$. The values were summed up for all the VOCs for obtaining the ozone production potential corresponding to each of the PMF derived factors for the daytime hours (07:00-18:00LT).

Secondary organic aerosol (SOA) potential was calculated for the PMF source factors using the literature SOA yields (Derwent et al., 2010) under low NO$_x$ conditions for benzene, toluene,

ethylbenzene, trimethylbenzene, styrene, methanol, isoprene, formaldehyde, acetaldehyde, acetone, formic acid and acetic acid using the equation given below for 07:00-18:00LT:

$$SOA\ potential = (\sum_i [VOC_i][SOAP_i]) \qquad (4)$$

**Reviewer comment:** The description of the methodology used to compare results of this study with the emission inventories estimates should be outlined.
**Author response:** done
**Changes in the manuscript:** We inserted a section 2.6 to describe this methodology which was earlier described in the results section (3.8) and have removed the method from section 3.8 to avoid repetition. Now section 3.8 only discusses the results.

**"2.6 Methodology for the comparison of PMF source factors with existing emission inventories**
Global Emission Database for Global Atmospheric Research (EDGARv4.3) inventory for the year 2012 (Huang et al. 2017), and two regional emission inventories: Regional Emission inventory in Asia (REAS v2.1) for the year 2008 (Kurokawa et al., 2013) and the Greenhouse Gas and Air Pollution Interactions and Synergies model (GAINS) (Amann et al., 2011) for the year 2010 (Stohl et al., 2015) were compared with our PMF output. The gridded inventory was filtered for Latitude: 27.4-34.9 N and Longitude: 72-79.8 E, i.e. the fetch region from which the air mass trajectories reach the receptor site within one day. This filtering is required because compounds with photochemical lifetimes of less than a day (e.g. styrene, C-8 and C-9 aromatics) feature prominently in several source profiles indicating that most of the transport sector emission were less than a day old when they reached the receptor site. Other compounds with longer lifetimes such as toluene (2 days), benzene (6 days) or acetonitrile (months) can reach the site from more distant sources. The wheat residue burning source shows the highest cross correlation with the regional fire counts for a lag time of 2 days indicating that emissions from distant sources can and do impact the site with a time lag. The chosen fetch region includes the areas where the maximum number of wheat residue burning fire counts are observed while avoiding a size that is too large to be consistent with the relatively unaltered signature of some of the other PMF source profiles.
Annual emissions were available for EDGAR (2012) and GAINS (2010), whereas, REAS provided monthly data (May 2008). However, Figure S5 shows that despite providing monthly data, the REAS emission inventory has very little seasonality for any of the sources. To facilitate the comparison of the PMF output of the month of May which is affected by a strongly seasonal source (crop residue burning) with emission inventories that provide only annual data as of now, we calculate hypothetical pie charts which attribute annual crop residue burning emissions over the region only to the 2.5 months when crop residue burning actually occurs (middle of October to end of November and May)."

**Reviewer comment:** 4. RESULTS AND DISCUSSIONS -
Overall results are too descriptive, and there are repetitions of information that figures already provide. I suggest to focus more on what can be deduced from the analysis rather than on its description.
**Author response:** We appreciate this advice by the anonymous reviewer #1 and have restructured our results and discussion section. The former section 3.1 has been combined with some details regarding the model output validation which were spread out over sections 3.2-3.6 and has been shifted to a new section "2.4 Validation of the PMF output" in response to one of the comments of the anonymous reviewer #2. Our Results and discussion section

now starts with the content of the former section 3.7 (now shifted to 3.1) "Split up of VOC Emission Sources in Mohali and their contribution to Ozone and SOA Formation Potential". Sections 3.2-3.6 containing the description of the PMF results for the individual factors have been re-written to focus on what the analysis means rather than on describing the results.

**Changes in the manuscript:**

**Section 3.2 now reads:**

[revised manuscript text omitted]

**Reviewer comment:** • Section 3.8 presents the comparison between the source apportionment study and emission inventories estimates, i.e. a point vs gridded data. Is it sufficient to filter gridded data for LAT LONG from which air mass trajectories reach the site within one day to make the comparison reliable?

**Author response:** Air is a rapidly moving medium, in particular in May when the average wind speed is 5.6 ms$^{-1}$. Hence, the comparison of a receptor point with a much larger gridded area of an emission inventory should not be a concern. In fact, Sofowote et al. 2015 (Atmos. Environ. 108:151–57) used the PMF to source apportion the impact of distant sources on the PM$_{2.5}$ aerosol burden at 5 remote locations in Ontario, Canada. We think that the more pertinent question is: How large should that gridded area be for a meaningful comparison? Many of the very specific tracers have short photochemical lifetimes of less than a day (e.g. styrene, C-8 and C-9 aromatics). Since these short lived compounds feature prominently in several source profiles, rather than being absent, this indicates that e.g. the 4-wheeler emissions on average have been subjected to photochemical aging for less than 4-10 hours prior to reaching the site. On the other hand, other compounds e.g. toluene (2 days), benzene (6 days) or acetonitrile (months) could have been transported much further away. The wheat residue burning source shows the greatest cross correlation for a lag time of 2 days indicating that emissions from distant sources can and do impact the site with a time lag. Hence we chose a compromise between the two sets of compounds in terms of lifetimes and delineated a fetch region of 1 day for the comparison with the emission inventories. This fetch region includes the areas where the maximum number of wheat residue burning fire counts are observed by satellites while avoiding a size that is too large to be consistent with the relatively unaltered signature of some of the other PMF source profiles.

**Changes in the manuscript:** We have inserted the following text into the newly created section 2.6

"This filtering is required because compounds with photochemical lifetimes of less than a day (e.g. styrene, C-8 and C-9 aromatics) feature prominently in several source profiles indicating that most of the transport sector emission were less than a day old when they reached the receptor site. Other compounds with longer lifetimes such as toluene (2 days), benzene (6 days) or acetonitrile (months) can reach the site from more distant sources. The wheat residue burning source shows the greatest cross correlation for a lag time of 2 days indicating that emissions from distant sources can and do impact the site with a time lag. The fetch region chosen for comparison with the emission inventories includes the areas where the maximum number of wheat residue burning fire counts are observed by satellites while

avoiding a size that is too large to be consistent with the relatively unaltered signature of some of the other PMF source profiles. "

**Reviewer comment:** Moreover, the study considers May 2012, while emissions inventory data are for 2008/2010. Which are the uncertainties in using these approaches in the comparison? Authors should justify and better describe these choices.

**Author response:** We have reduced the uncertainties of the comparison by switching from EDGARv4.2 to the more recent version 4.3.2 for the year 2012. As far as REASv2.1 for the year 2008 is concerned, we could not improve the comparison as the NMVOC dataset of the MIX Asia 2010 inventory is identical to the NMVOC dataset of the REAS 2008 inventory. When it comes to the uncertainties introduced by comparing one month's data with an annual average emission inventory is concerned there are two parts to the answer.

1)The first part of the answer is that at present the only inventory that gives monthly data is in no way better than the inventories which provide only annual average data as the monthly data hardly differs from the sum of annual emissions divided by 12. Methane emissions from rice paddies in Punjab persist in the REAS emission inventory throughout the year even in months in which rice is not grown. Other sources do not appear to have been treated differently. Hence de facto there is no seasonality in any of the emission inventories available at present, a short coming that must be overcome in the long run but is beyond the scope of this work.

2) For emission inventories that do not provide monthly data, we have facilitated the comparison of the PMF output of the month of May which is affected by a strongly seasonal source (crop residue burning). To do so, we calculate hypothetical pie charts which attribute annual crop residue burning emissions over the region only to the 2.5 months when crop residue burning actually occurs (middle of October to end of November and May). This should reduce the uncertainty of the comparison. It allows to assess whether the model has the correct annual total emissions of the crop residue burning source and just lacks the proper distribution in the form of monthly data or is off with respect to the total annual emissions itself.

**Changes in the manuscript:** The following two text segments have been included in section 2.6

"Annual emissions were available for EDGAR (2012) and GAINS (2010), whereas, REAS provided monthly data (May 2008). However, Figure S6 shows that despite providing monthly data, the REAS emission inventory has very little seasonality for any of the sources."

"To facilitate the comparison of the PMF output of the month of May which is affected by a strongly seasonal source (crop residue burning) with emission inventories that provide only annual data, we calculate hypothetical pie charts which attribute annual crop residue burning emissions over the region only to the 2.5 months when crop residue burning actually occurs (middle of October to end of November and May)."

Figure 8 has been changed – so has the accompanying text.

[Figure]

Figure 8 has been revised and now includes EDGAR v4.3.2 (2012) instead of v4.2 (2008) and have updated the discussion accordingly. The latest EDGAR represents a significant improvement over the EDGAR HTAP and v4.2.

We have also added supplementary figures to compare speciated emission inventories with the PMF output for individual aromatic compounds

[Figure]

**Figure S8a:** Comparison of the PMF output with benzene emission inventories for the study region.

[Figure]

**Figure S8b:** Comparison of the PMF output with toluene emission inventories for the study region.

[Figure]

**Figure S8c:** Comparison of the PMF output with xylenes in the emission inventories for the study region.

[Figure]

**Figure S8d:** Comparison of the PMF output of C-9 aromatic compounds with the class "other aromatic compounds" in the emission inventories for the study region.

**Reviewer comment:** 5. CONCLUSIONS - It would be more valuable for the reader if the authors focused more on the achievements and implications of the results. The last paragraph of 3.8 may be included in the conclusions rather than in results.

Done we have shifted the paragraph and have re-written the conclusions. It now reads as follows:

"Our results highlight that for accurate air quality forecasting and modelling it is essential that emissions are attributed only to the months in which the activity actually occurs. This is important both for emissions from crop residue burning (which occur in May and from Mid-October to the end of November). Annually averaged emissions are unlikely to yield accurate air quality forecast in regions affected by such seasonal events. At present, more specialized fire emission inventories such as FINN (Wiedinmyer et al., 2011) must be used to account for the full seasonality and day to day variations of open burning emissions. We also demonstrate, that the source profiles obtained as PMF output can be validated and matched against samples collected at the potential sources to validate the factor identification.

For the human class I carcinogen benzene, the traffic factor alone contributed to 47% of the total benzene mass at this receptor site followed by residential biofuel use and waste disposal (25%) and industrial emissions and solvent use (20%). This stands in stark contrast to various emission inventories which consider domestic biofuel usage (39%), agricultural residue burning (19%) and industries (24%) to be the most important sources of benzene emissions. Since the annual NAAQS for benzene is exceeded at this receptor site (Chandra and Sinha, 2016), all three sectors must be targeted for emission reductions.

For the emerging contaminant isocyanic acid, photochemical formation from precursors (37%), wheat residue burning (25%) and biofuel usage and waste disposal (18%) were the largest contributors to human exposure. The monthly average isocyanic mixing ratio of 1.4 ppb exceeds concentrations that can, after dissociation at blood pH, result in blood cyanate ion concentrations (Roberts et al., 2011) high enough to produce significant health effects in humans (Wang et al., 2007) such as atherosclerosis, cataracts and rheumatoid arthritis due to

protein damage. Peak mixing ratios of this compound exceed 3 ppb in some night time wheat residue burning plumes. Wheat residue burning was also the single largest source of the photochemical precursors of isocyanic acid, namely, formamide, acetamide and propanamide, indicating that this source must be most urgently targeted to reduce human concentration exposure to isocyanic acid.

Overall it appears that none of the emission inventories is ideal at the present. Our PMF solution suggests that transport sector emissions may be underestimated by GAINSv5.0 and EDGARv4.3.2, while the combined effect of residential biofuel use and waste disposal emissions as well as the VOC burden associated with solvent use may be overestimated by all emission inventories. Agricultural waste burning emissions of some of the detected compound groups (ketones, aldehydes and acids) are missing in the EDGARv4.3.2 inventory while aromatic emissions from the same source appear to be overestimated. Thus, large improvements are required in existing emission inventories for correct source attribution and inclusion of missing compounds over this densely populated region of the world."

**3 Minor comments**
**Reviewer comment:** 1. First author name (Pallavi) is missing.
Pallavi is a single name author. Her orcid is https://orcid.org/0000-0003-3664-6260

**Reviewer comment:** 2. Page 2 line 5 '...deserve further study' this sentence need citation.
**Author response:** This sentence refers to the previous sentence. Citations have been added to the previous sentence (Pawar et al. 2015, Sinha et al. 2014, Kumar et al. 2016)

**Reviewer comment:** 3. Page 2 line 31 '...and strong photochemistry' this sentence need citation.
**Author response & changes in the manuscript:** A citation to Sinha et al. 2014 has been added

**Reviewer comment:** 4. Section 2.3: need to add cross references to Table S3, Figure S4 a, b c.
**Author response:** done, we have added the cross reference in line 26 page 3
Figures S4 a, b c show how the factor profile, percentage of each VOC originating from a certain source, and the factor contribution change while increasing the number of factors in the model.
and line 1 page 4

A list of the constraints applied is provided in the supplementary table S3

**Reviewer comment:** 5. Page 9 line 16 'However, Figure S5..'. It is Figure S6 in the Supplement.
**Author response:** we have changed the numbering of several figures in the supplement as Reviewer #2 asked us to include an additional plot. The numbers are now consistent with the numbering in the manuscript.
**Reviewer comment:** 6. Figure 1 (b): add lat - long grid. It may be worth to add in the caption the exact coordinates of the site.
**Author response:** We have added the exact coordinates of the site instead.
We don't agree that adding a grid to the bottom figure is a good idea. It becomes a mess since Google Earth does not seem to allow us to define the grid spacing. It doesn't even seem to allow us to choose a different font size for the location labels and the grid labels. We are dealing with an area of less than 1 x 1 degree, so the figure with grid on looks ugly.
Figure 1b with grid on:

[Figure]

**Other minor corrections:** While preparing the new supplementary Figure S7 a small mistake in the calculation of the factor time series in μg/m$^3$ was spotted and corrected in Figure 5,7, S5c and throughout the manuscript.